EMBO
Molecular Medicine

# Excess hydrogen sulfide and polysulfides production underlies a schizophrenia pathophysiology

Masayuki Ide[1,2,†], Tetsuo Ohnishi[1,†], Manabu Toyoshima[1], Shabeesh Balan[1] (ID), Motoko Maekawa[1], Chie Shimamoto-Mitsuyama[1], Yoshimi Iwayama[1,3], Hisako Ohba[1], Akiko Watanabe[1], Takashi Ishii[4], Norihiro Shibuya[5,6], Yuka Kimura[5,6], Yasuko Hisano[1], Yui Murata[7], Tomonori Hara[1,8], Momo Morikawa[9], Kenji Hashimoto[10], Yayoi Nozaki[1], Tomoko Toyota[1], Yuina Wada[1,11], Yosuke Tanaka[9], Tadafumi Kato[12], Akinori Nishi[13], Shigeyoshi Fujisawa[14], Hideyuki Okano[15], Masanari Itokawa[16], Nobutaka Hirokawa[9] (ID), Yasuto Kunii[17,18], Akiyoshi Kakita[19], Hirooki Yabe[17], Kazuya Iwamoto[7], Kohji Meno[4], Takuya Katagiri[20], Brian Dean[21,22], Kazuhiko Uchida[23], Hideo Kimura[5,6] & Takeo Yoshikawa[1,*] (ID)

## Abstract

Mice with the C3H background show greater behavioral propensity for schizophrenia, including lower prepulse inhibition (PPI), than C57BL/6 (B6) mice. To characterize as-yet-unknown pathophysiologies of schizophrenia, we undertook proteomics analysis of the brain in these strains, and detected elevated levels of Mpst, a hydrogen sulfide ($H_2S$)/polysulfide-producing enzyme, and greater sulfide deposition in C3H than B6 mice. *Mpst*-deficient mice exhibited improved PPI with reduced storage sulfide levels, while *Mpst*-transgenic (Tg) mice showed deteriorated PPI, suggesting that "sulfide stress" may be linked to PPI impairment. Analysis of human samples demonstrated that the $H_2S$/polysulfides production system is upregulated in schizophrenia. Mechanistically, the *Mpst*-Tg brain revealed dampened energy metabolism, while maternal immune activation model mice showed upregulation of genes for $H_2S$/polysulfides production along with typical antioxidative genes, partly via epigenetic modifications. These results suggest that inflammatory/oxidative insults in early brain development result in upregulated $H_2S$/polysulfides production as an antioxidative response, which in turn cause deficits in bioenergetic processes. Collectively, this study presents a novel aspect of the neurodevelopmental theory for schizophrenia, unraveling a role of excess $H_2S$/polysulfides production.

**Keywords** energy metabolism; epigenetics; hydrogen sulfide and polysulfides; prepulse inhibition; proteomics

**Subject Categories** Chromatin, Transcription & Genomics; Neuroscience

See also: **M Simonneau** (December 2019)

---

1. Laboratory of Molecular Psychiatry, RIKEN Center for Brain Science, Wako, Saitama, Japan
2. Department of Psychiatry, Division of Clinical Medicine, Faculty of Medicine, University of Tsukuba, Tsukuba, Ibaraki, Japan
3. Support Unit for Bio-Material Analysis, Research Division, RIKEN Center for Brain Science, Wako, Saitama, Japan
4. Research& Development Department, MCBI Inc, Tsukuba, Ibaraki, Japan
5. Department of Pharmacology, Sanyo-Onoda City University, Sanyo-Onoda, Yamaguchi, Japan
6. Department of Molecular Pharmacology, National Institute of Neuroscience, National Center of Neurology and Psychiatry, Kodaira, Tokyo, Japan
7. Department of Molecular Brain Science, Graduate School of Medical Sciences, Kumamoto University, Kumamoto, Japan
8. Department of Organ Anatomy, Tohoku University Graduate School of Medicine, Sendai, Miyagi, Japan
9. Department of Cell Biology and Anatomy, Graduate School of Medicine, The University of Tokyo, Tokyo, Japan
10. Division of Clinical Neuroscience, Chiba University Center for Forensic Mental Health, Chiba, Japan
11. Graduate School of Humanities and Sciences, Ochanomizu University, Tokyo, Japan
12. Laboratory for Molecular Dynamics of Mental Disorders, RIKEN Center for Brain Science, Wako, Saitama, Japan
13. Department of Pharmacology, Kurume University School of Medicine, Kurume, Fukuoka, Japan
14. Laboratory for Systems Neurophysiology, RIKEN Center for Brain Science, Wako, Saitama, Japan
15. Department of Physiology, Keio University School of Medicine, Tokyo, Japan
16. Center for Medical Cooperation, Tokyo Metropolitan Institute of Medical Science, Tokyo, Japan
17. Department of Neuropsychiatry, School of Medicine, Fukushima Medical University, Fukushima, Japan
18. Department of Psychiatry, Aizu Medical Center, Fukushima Medical University, Aizuwakamatsu, Fukushima, Japan
19. Department of Pathology, Brain Research Institute, Niigata University, Niigata, Japan
20. Department of Pharmacy, Faculty of Pharmacy, Iryo Sosei University, Iwaki, Fukushima, Japan
21. The Florey Institute of Neuroscience and Mental Health, Howard Florey Laboratories, The University of Melbourne, Parkville, Vic., Australia
22. The Centre for Mental Health, Swinburne University, Hawthorn, Vic., Australia
23. Department of Molecular Oncology, Division of Biomedical Science, Faculty of Medicine, University of Tsukuba, Tsukuba, Ibaraki, Japan

*Corresponding author. Tel: +81 48 467 5968; Fax: +81 48 467 7462; E-mail: takeo.yoshikawa@riken.jp
†These authors contributed equally to this work

---

# Introduction

Schizophrenia is a severe mental illness featuring three major symptomatic domains: positive symptoms (hallucinations, delusions, etc.), negative symptoms (affective flattening, avolition, etc.), and cognitive deficits (disorganized thought, etc.) (American Psychiatric Association, 2013). This illness exhibits a life-time prevalence of approximately one percent worldwide. Repeated relapses of psychotic symptoms often lead to a deterioration of brain function, and eventually to end-stage illness in some cases, characterized by persistent symptoms and profound functional disabilities (Lewis & Lieberman, 2000). Though causal mechanisms are elusive, accumulating lines of evidence have shown abnormalities in early neurodevelopment processes, stemmed from genetic aberrations and environmental factors, such as maternal immune activation, for the etiopathogenesis of schizophrenia (neurodevelopmental hypothesis) (Knuesel et al, 2014; Estes & McAllister, 2016; Birnbaum & Weinberger, 2017). And for symptomatic treatments, drugs targeting dopaminergic systems are predominantly used (Murray et al, 2017). However, because currently available therapeutics has limitations in terms of efficacies and adverse effects, a new paradigm is needed for the development of novel drugs.

Here, we hypothesize that examination of the traits and associated molecular underpinnings in inbred mouse strains could potentially identify as-yet-unknown pathophysiologies of schizophrenia. Pursuing this hypothesis, we have already reported that across 4 strains of mice, C57BL/6N (B6) mice exhibited the highest prepulse inhibition (PPI) scores while C3H/HeN (C3H) the lowest (Watanabe et al, 2007). PPI is the normal suppression of a startle response when a low-intensity stimulus immediately precedes an unexpected stronger startling stimulus. As a reproducible phenotypic marker, impaired PPI reflects sensorimotor gating deficits, and is typically regarded as an endophenotype for schizophrenia (Braff et al, 2001; Roussos et al, 2016).

To explore the molecular signature underlying the behavioral differences between B6 and C3H, we performed proteomic analyses, using 2D-DIGE (two-dimensional difference gel electrophoresis) and MALDI-TOF MS (matrix-assisted laser desorption/ionization time-of-flight mass spectrometry). This screening step revealed an increase in the levels of a hydrogen sulfide ($H_2S$)- and polysulfides- ($H_2S$/polysulfides)-producing enzyme, Mpst (3-mercaptopyruvate sulfurtransferase; also known as 3MST) (Appendix Fig S1), in C3H mice compared to B6 animals. Then, we comprehensively assessed the biological possibility of the novel theory of "excessive $H_2S$/polysulfides production" in schizophrenia, and pursued the mechanism underlying the functional consequence and causative origin of this phenomenon.

# Results

## Proteomic analyses of brain and splenic lymphocytes from B6 and C3H mice identified Mpst

We performed proteomic analyses using brain and lymphocyte preparations from B6 and C3H mice to detect a homologous biomarker in the peripheral blood of schizophrenia patients. As shown in Fig 1A and B, and Appendix Tables S1 and S2, of the 1,093 spots identified from the brain using 2D-DIGE, 43 showed significant differences in expression between B6 and C3H animals, with our criteria of a fold change > 1.2 and $P < 0.05$. Of the 1,400 spots detected from the lymphocyte preparations, 131 showed significant differences in expression between B6 and C3H mice. Sixteen of these differentially expressed protein spots showed consistent change trends between the tissues from the two mouse strains (Appendix Table S2).

We successfully identified the molecular entities of nine of these protein spots, by peptide mass fingerprinting (PMF), and found five different proteins: heat shock 70 kDa protein 9 (mortalin, Hspa9), nucleophosmin/nucleoplasmin (Npm1), Mpst, peroxiredoxin 6 (Prdx6), and nucleoside diphosphate kinase B (Nme2) (Appendix Tables S3 and S4). The proteins Hspa9, Npm1, Prdx6, and Nme2 appeared as paired spots at different isoelectric points (pI) (Fig 1C, D, F and G). Sequencing of the genes encoding these proteins revealed that each gene harbored single-nucleotide polymorphisms (SNPs) or insertion/deletion polymorphisms, which altered the amino acid sequences between the two mouse strains (Appendix Table S5 and Appendix Fig S2). The theoretical pI and molecular mass values calculated based on DNA sequences and possible posttranslational modifications were consistent with the observed spot profiles of these proteins in 2D gels, and were confirmed by 2D Western blotting (Fig 2A–J). By using 2D-DIGE, Mpst appeared as a single spot that was expressed in only C3H mice (Fig 1E). Two-dimensional Western blotting confirmed this spot for C3H animals (Fig 2F) and detected a faint spot for B6 mice at a different pI from the spot observed for C3H mice (Fig 2E). Sequencing of the Mpst gene revealed a Asp102Gly polymorphism (Appendix Table S5 and Appendix Fig S2), which can explain the differential mobility in the 2D gel between the two mouse strains. The Mpst protein catalyzes the transfer of a sulfur ion from 3-mercaptopyruvate to cyanide or other thiol compounds (Szabo, 2007; Kimura, 2015; Wallace & Wang, 2015), and this reaction produces $H_2S$/polysulfides and detoxifies cyanide (Appendix Fig S1). The Asp102Gly polymorphism is predicted to have little effect ("benign") by the PolyPhen-2 algorithm (http://genetics.bwh.harvard.edu/pph2/). Indeed, the functional assay conducted by preparing Asp102 and Gly102 Mpst constructs showed no significant differences in enzymatic activity between the variants (Appendix Fig S3 and Appendix Table S6).

The Mpst spot was the only protein to show differential expression, exhibiting lower expression in B6 mice than in C3H mice. The protein expression levels for Mpst were confirmed by standard Western blot analyses of B6 and C3H mice using both brains and splenic lymphocytes: significantly higher expression of Mpst was observed in the frontal cortex of the C3H mouse brain than in that of the B6 brain using both anti-Mpst N-terminus (Mpst-N, $P = 0.03$, Fig EV1A) and anti-Mpst C-terminus (Mpst-C, $P = 0.02$, Fig 3A) antibodies, though only a marginally increased expression of Mpst was observed in C3H mice in splenic lymphocytes (Mpst-N, $P = 0.25$; Mpst-C, $P = 0.11$) (Figs 3B and EV1B). Because of the nonsignificant differences in protein expression levels in lymphocytes between the two strains, we hereafter focused on mainly brain tissues.

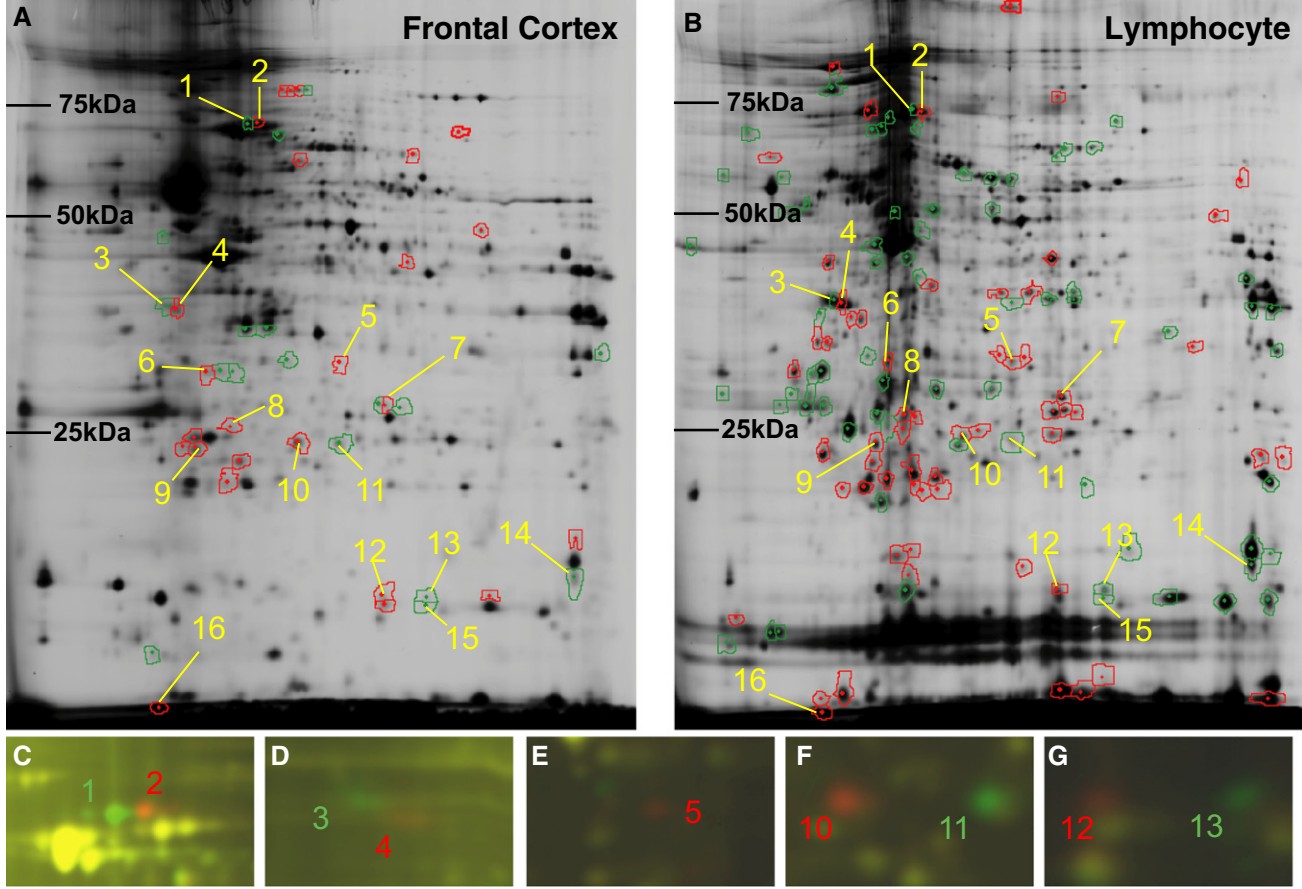

**Figure 1. Differential proteomic analysis of brain and lymphocyte tissues from B6 and C3H mice using 2D-DIGE.**

A, B  Results of 2D-DIGE on frontal cortex tissue (A) and lymphocytes (B). Green and red rings denote spots showing significantly increased expression in B6 and C3H mice, respectively.

C–G  Merged images of Cy3 (green) and Cy5 (red): green and red spots represent significantly increased expression of the corresponding protein isoforms in B6 and C3H mice, respectively. Nine (spot nos. 1, 2, 3, 4, 5, 10, 11, 12 and 13) out of the 16 spots were successfully identified as mortalin (Hspa9) (C), nucleophosmin (Npm1) (D), mercaptopyruvate sulfurtransferase (Mpst) (E), peroxiredoxin 6 (Prdx6) (F) and nucleoside diphosphate kinase B (Nme2) (G).

Data information: Significance for differential expression between B6 and C3H was defined as $P$ value of < 0.05 (unpaired two-tailed $t$-test) and fold change > 1.2. Yellow numbers indicate the 16 spots that showed consistent alterations between brain and lymphocyte samples from the two mouse strains.

Source data are available online for this figure.

## Sulfide content in the brain was higher in C3H than in B6 mice

Next, we examined whether upregulation of Mpst caused elevated deposition of the free form of sulfur, i.e., $H_2S$, and/or stored sulfur, i.e., acid-labile sulfur and bound sulfane sulfur, which includes $H_2S_n$ ($n = 2$: persulfide, $n > 2$: polysulfides) and persulfurated-cysteine, -glutathione (GSH), and -proteins (Appendix Fig S4) (Kimura, 2015). $H_2S_2$ and $H_2S_3$ were recently shown to be generated from 3-mercaptopyruvate by Mpst and to be present in the brain (Kimura *et al*, 2013, 2015, 2017; Koike *et al*, 2017; Nagahara *et al*, 2018). Acid-labile sulfur is mainly found in an iron–sulfur complex attached to enzymes belonging to the respiratory chain including aconitase (Ishigami *et al*, 2009) (Appendix Fig S4). This complex releases $H_2S$ under acidic conditions, but not under physiological conditions. Bound sulfane sulfur releases $H_2S$ under reducing conditions (Appendix Fig S4) (Ishigami *et al*, 2009). As shown in Fig 3D and E, the $H_2S$ and acid-labile sulfur content was

unchanged between the two mouse strains, but the bound sulfane sulfur content was markedly elevated in the C3H brain compared to the B6 brain (Fig 3F), suggesting an increase in the levels of $H_2S_n$ and per- and poly-sulfurated molecules. The endogenous $H_2S$ produced by enzymes may be readily oxidized to $H_2S_n$ and incorporated into bound sulfane sulfur (Shibuya *et al*, 2009; Kimura *et al*, 2015).

### *Mpst* deficiency or overexpression, and external sulfides affect mouse behaviors

Since Mpst was observed to be overexpressed in C3H when compared to that in B6, it is important to determine whether Mpst plays a role in the distinct PPI levels between B6 and C3H. To assess the role of Mpst, we generated *Mpst* knockout (KO) mice in the C3H background (Appendix Fig S5), and *Mpst*-transgenic (Tg) mice in the B6 background (Appendix Fig S6). The *Mpst*-KO mice showed

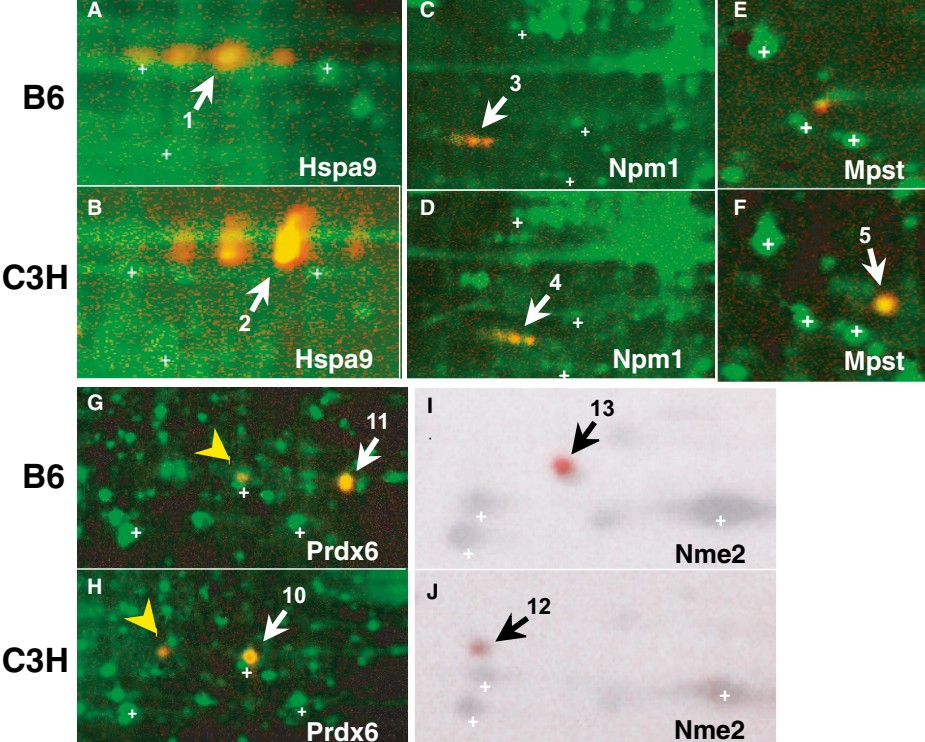

**Figure 2. Identified proteins visualized by 2D Western blotting.**

A–J  Whole protein extracts from brain tissue of B6 (A, E, G, I). Whole protein extracts from brain tissue of C3H (B, F, H, J). Whole protein extracts from lymphocytes of B6 (C) and C3H (D). Npm3 expression levels were low in the brain (C, D). Hspa9 (mortalin) (A, B), Npm1 (nucleophosmin) (C, D), Mpst (mercaptopyruvate sulfurtransferase) (Mpst) (E, F), Prdx6 (peroxiredoxin 6) (G, H) and Nme2 (nucleoside diphosphate kinase B) (I, J) were detected by 2D Western blotting using the corresponding antibodies and chemiluminescence (red). The chemiluminescent signal of Nme2 was visualized by the LAS 3000 chemiluminescence image analyzer and the other signals were visualized by a Typhoon 9400.

Data information: White crosses (+) indicate landmark spots. Spot numbers (indicated by arrows) correspond to the spot numbers in Fig 1. Yellow arrowheads (G, H) indicate the overoxidized form of Prdx6.

elevated PPI at a prepulse level of 78 dB compared to their wild-type (WT) littermates (Fig 4A). Interestingly, C3H inbred mice exhibited an enhanced acoustic startle response (ASR) compared to B6 (Fig 4B), which has been previously reported in schizophrenic model mice (Egashira *et al*, 2005; Duncan *et al*, 2006). *Mpst*-KO mice in the C3H background showed diminished ASR relative to their WT littermates (Fig 4C). In contrast, the *Mpst*-Tg mice exhibited decreased PPI at prepulse levels of 78 dB and 82 dB (Fig 4D) and magnified ASR compared to the non-Tg littermates (Fig 4E). These results indicate that upregulation of the *Mpst* causes schizophrenia-related impaired PPI and exaggerated ASR.

Regarding the sulfide levels in the brain, $H_2S$ levels were unaltered by the change in copy number of *Mpst* (Fig 4F and G), and the levels of acid-labile sulfur were slightly decreased in the *Mpst*-KO mice compared to the WT littermates (Fig 4H). Notably, the *Mpst*-KO mice showed a dramatic decrease in bound sulfane sulfur levels compared to the WT mice (Fig 4I). These results indicate that when the level of Mpst is decreased, sulfide deposition in the brain is reduced. The *Mpst*-Tg mice did not show any differences in the levels of both free and stored forms of sulfur (Fig 4G, J and K), probably because the magnitude of transgene expression was not sufficiently large

(Appendix Fig S6) for the detection of distinct levels of sulfides in our assay system.

Next, we examined whether external administration of the $H_2S$-producing agent, NaHS, can elicit an impaired PPI. Chronic injections of NaHS decreased the PPI of C3H mice, although that of B6 mice was unchanged (Fig EV2). B6 mice may be relatively resistant to excess $H_2S$ and polysulfides (also see Fig 4K).

### Expression of the $H_2S$ synthesis system is upregulated in schizophrenia

The results thus far demonstrated that upregulation of Mpst and concomitant accumulation of sulfides in the brain possibly causes the impairment of PPI, a representative biological trait of schizophrenia. There are two other well-known enzymes, namely, Cbs/CBS (cystathionine-beta-synthase) and Cth/CTH (cystathionine gamma-lyase), that are also involved in the production of $H_2S$ (Appendix Fig S1) (Szabo, 2007; Kimura, 2015; Wallace & Wang, 2015). Real-time quantitative PCR (RT–qPCR) analyses revealed that the levels of the *Mpst* and *Cth* mRNAs increased in C3H mouse brains (Fig 3C). The absolute expression levels of the three genes in the mouse brain measured by digital RT–PCR showed the trend

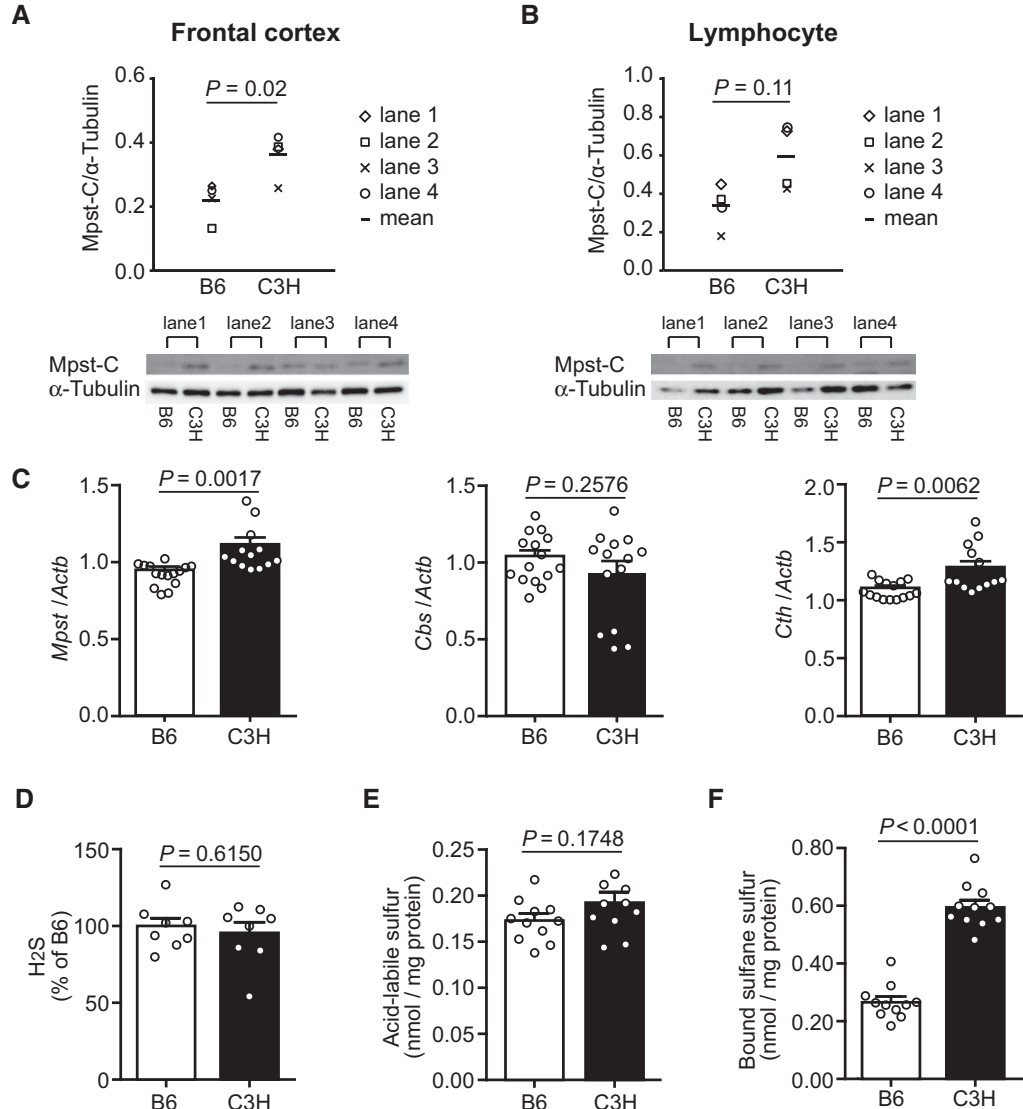

**Figure 3. Expression of Mpst/*Mpst* and genes encoding the other H₂S-producing enzymes and profile of H₂S metabolic states in mice.**

A, B   Mpst protein levels in the brain and lymphocytes from B6 and C3H mice were quantified by standard Western blotting with anti-C-terminal Mpst antibodies (for anti-N-terminal Mpst antibodies, see Fig EV1). The expression levels of Mpst were normalized using α-tubulin. Bar graphs show the mean expression levels of Mpst in brain (A) and lymphocyte (B) tissues.

C   Transcript expression levels of genes encoding three H₂S-producing enzymes in the frontal cortex of B6 ($n$ = 15) and C3H mice ($n$ = 14). The values represent the mean ± SEM.

D   H₂S content in the frontal cortex of B6 ($n$ = 8) and C3H mice ($n$ = 8). The values are relative to those of B6 and represent mean ± SEM.

E   Levels of labile sulfur in the frontal cortex of B6 ($n$ = 10) and C3H mice ($n$ = 10). The values are relative to those of B6 and represent mean ± SEM.

F   Levels of bound sulfane sulfur in the frontal cortex of B6 ($n$ = 10) and C3H mice ($n$ = 10). The values are relative to those of B6 and represent mean ± SEM.

Data information: $P$ values were calculated using unpaired two-tailed $t$-test.
Source data are available online for this figure.

*Mpst ~ Cbs > Cth* (Fig EV3). Therefore, the higher sulfide levels in C3H than in B6 brains mainly stemmed from differential *Mpst* expressions between the two strains. Interestingly, the expression levels of the differentially expressed *Mpst* and *Cth* were positively correlated with each other (Fig 5A), indicating concerted operation of the H₂S-producing system.

In human postmortem brains [Brodmann area (BA) 8, frontal cortex] (1st sample set) (Appendix Table S7), RT–qPCR analyses revealed increased expressions of *MPST* and *CBS* in subjects with schizophrenia (Fig 6A and B), with no change in *CTH* mRNA levels (Fig 6C). The absolute expression levels of the three genes in the human brain followed the trend *CBS > CTH > MPST* (Fig EV3), suggesting a relatively strong role of CBS in the regulation of the total sulfide levels in the human brain. The expression levels of differentially expressed *CBS* and *MPST* were well correlated with each other in both the control and schizophrenia brain

samples (Fig 5B). As in the case of C3H mice, $H_2S$/polysulfides production is thought to be upregulated by coordinated leveraging of multiple genes for $H_2S$/polysulfides synthesis in the schizophrenia brain samples.

To verify the idea of upregulated $H_2S$/polysulfides production state in schizophrenia, we examined a different postmortem brain sample set (BA17) ($2^{nd}$ sample set) (Ohnishi et al, 2019) (Appendix Table S8) and evaluated phenotypic features of schizophrenia that are associated with elevated $H_2S$ production system. In this sample set, the MPST protein expression levels were higher in the schizophrenia than in the control groups (Fig 7A). Interestingly, the MPST levels in schizophrenia were positively correlated with symptom severity scores (a sum of positive symptom score, negative symptom score, and general score) at 3 months prior to the death rated by using the Diagnostic Instrument for Brain Studies (DIBS) (Ohnishi et al, 2019) (Fig 7B). The results suggest that patients with schizophrenia under "sulfide stress" manifest relatively severe psychotic symptoms. Here it was difficult to precisely measure $H_2S$/polysulfides levels in human postmortem brains, because for an accurate measurement, samples should have been quickly removed and frozen immediately after death.

We also examined reprogrammed neuronal lineage cells from patients with schizophrenia. Neurospheres are composed of free-floating clusters of neural stem or progenitor cells, differentiated from induced pluripotent stem cells (iPSCs) (Maekawa et al, 2015; Toyoshima et al, 2016). iPSCs-derived neurospheres from schizophrenia patients carrying 22q11.2 micro-deletions (Fig 6D) (Bundo et al, 2014; Maekawa et al, 2015; Toyoshima et al, 2016) showed a significant increase in CBS mRNA compared to those derived from control subjects (Fig 6E). Absolute expression levels of CBS are much higher in early developmental cells, such as iPSCs and iPSCs-derived neurospheres, than in terminally differentiated cells (postmortem brain cells, hair follicles, peripheral blood cells (PBCs), and mouse brain) (Fig EV3). The results indicate that the $H_2S$ production system might be already elevated from the early neurodevelopmental stage in schizophrenia, which is associated with "neurodevelopmental theory" (Rapoport et al, 2012; Birnbaum & Weinberger, 2017).

We next addressed whether $H_2S$/polysulfides generation could be a biomarker of schizophrenia. The absolute transcript expression levels of the three genes for $H_2S$/polysulfides production in PBCs were all very low, but MPST exhibited the highest expression levels (Fig EV3). However, the MPST levels in PBCs exhibited no significant changes between subjects with schizophrenia and control (Fig 6F, Appendix Table S9), and plasma MPST protein levels were also unchanged between the two groups (Fig 6G and Appendix Table S9). These results seem to correspond to the mouse data that in mouse lymphocytes Mpst expression levels were not different between B6 and C3H, and suggest that it would be difficult to identify a biomarker from peripheral blood. The large proportion of Mpst protein in blood sample exists in red blood cells (RBCs) (Włodek & Ostrowski, 1982). Because RBCs do not have nuclei, the levels of Mpst in plasma may not be affected by differential gene expression. Then, we examined hair follicles as a source of biomarker genes ($n = 166$ for control and $n = 149$ for schizophrenia) (Maekawa et al, 2015). The MPST was the only substantially expressed transcript

among the three genes for $H_2S$ synthesis in this mini-organ (Figs 6H and EV3) (Maekawa et al, 2019), and MPST mRNA expression was increased in subjects with schizophrenia (Fig 6I, Appendix Table S10). None of the confounding factors, such as age at examination, BMI (body mass index), anti-psychotic drug dose, age of onset of schizophrenia, duration of the illness, and smoking, were significantly correlated with MPST mRNA expression in scalp hair follicles (Appendix Fig S7). Receiver operating characteristics (ROC) curve analysis determined an optimal cut-off level of 0.876 based on the maximum Youden index. With this cut-off level for the MPST/GAPDH mRNA ratio, the sensitivity and specificity were 73.6 and 47.2%, respectively (area under the curve = 0.602) (Fig 7C). The results suggest that the subset of schizophrenia with "sulfide stress" pathophysiology could be identified using this surrogate marker.

## Excess $H_2S$/polysulfides production elicits dampened expression of genes for energy metabolism, and impairs mitochondrial energy metabolism and diminishes spine density

To identify potential convergent pathways relevant for functional impairments elicited by excessive $H_2S$/polysulfides production in the brain, we performed RNA-seq analyses using Mpst-KO and Tg mice. Upon disruption of Mpst expression in the frontal cortex, a total of 480 genes were significantly dysregulated (239 upregulated and 241 downregulated) in the KO mice compared to the WT littermates ($P < 0.05$) (Fig 8A and Appendix Table S11). Gene ontology analysis revealed that the dysregulated genes were mainly enriched in biological processes involved in mRNA splicing and processing (Appendix Table S12). Transcriptomic analysis of the frontal cortex revealed significant dysregulation of 1,136 genes (519 upregulated and 617 downregulated) in the Mpst-Tg mice compared to the non-Tg littermates ($P < 0.05$) (Fig 8B, Appendix Table S13). Interestingly, the dysregulated genes in the Mpst-Tg mice were significantly enriched in the glutamatergic synaptic transmission and synaptic signaling-related ontology terms, along with the cellular metabolic processes (Fig 8C and Appendix Table S12). The roles of synaptic dysregulation and glutamatergic signaling impairments in the pathogenesis of schizophrenia are well-known (Uno & Coyle, 2019). Interestingly, we have also previously showed that deficits in Fabp7, a critical gene for regulating PPI, affect neurogenesis, NMDA signaling (Watanabe et al, 2007), and glial cell integrity (Ebrahimi et al, 2016), thus suggesting that genetic network for the PPI endophenotype overlaps with that of neurodevelopment and synaptic properties. In addition, Ingenuity Pathway Analysis™ demonstrated significant enrichment of molecular pathways, e.g., glycolysis I, TCA (tricarboxylic acid) cycle II, and dopamine-DARPP32 feedback in cAMP signaling, for the downregulated genes ($P < 0.05$) (Fig 8D and Appendix Table S14). This evidence is consistent with energy metabolism dysfunction (Zuccoli et al, 2017; Sullivan et al, 2019) and the involvement of deficits in DARP32 systems in the pathogenesis of schizophrenia (Kunii et al, 2014; Wang et al, 2017; Zuccoli et al, 2017).

We specifically examined the expression levels of Pvalb, because impairments of parvalbumin (Pvalb, PV)-containing inhibitory GABA ($\gamma$-aminobutyric acid) neurons in the dorsolateral prefrontal cortex is a well-recognized finding in schizophrenia

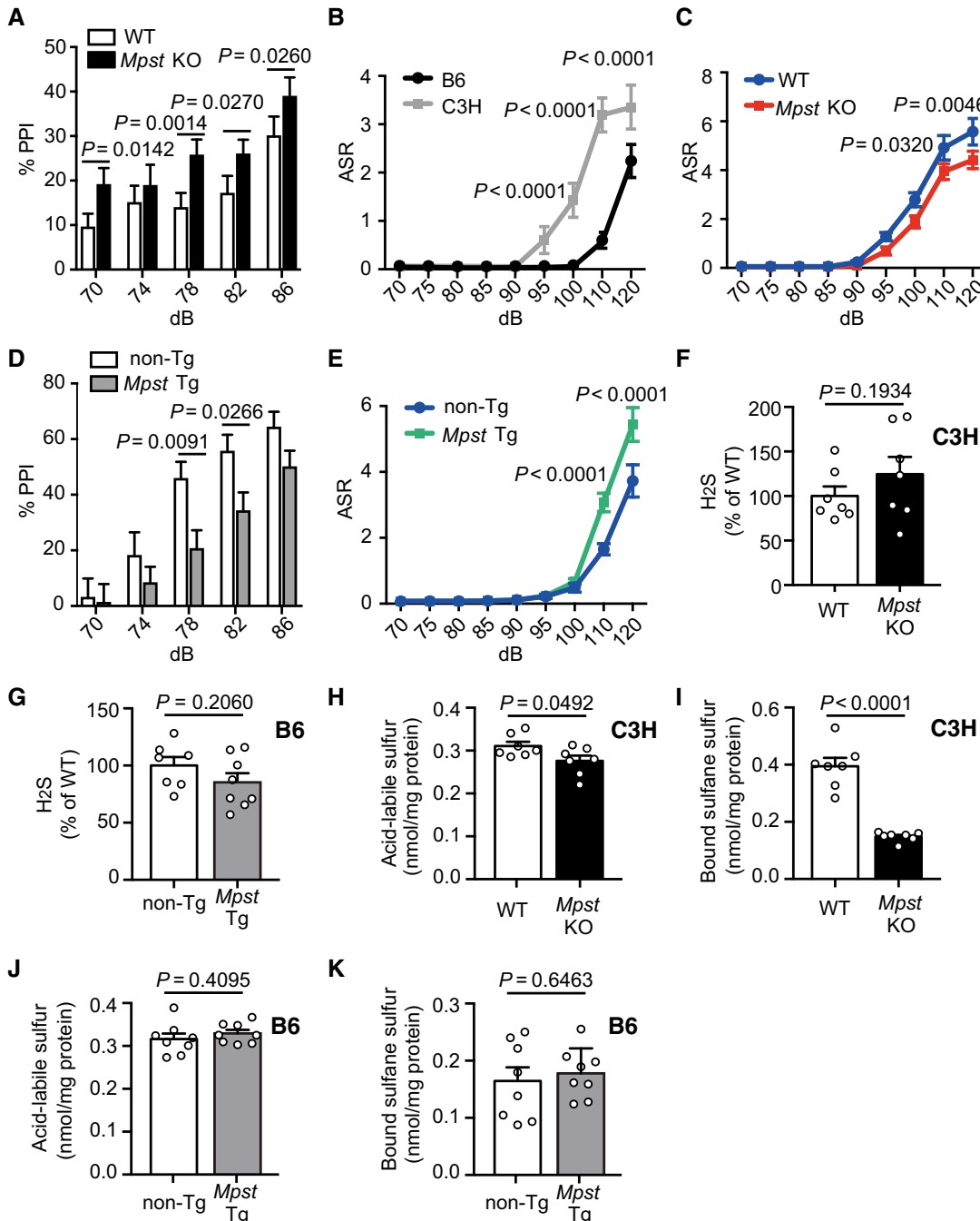

**Figure 4. Mouse behaviors and sulfide deposition in *Mpst* knockout (KO) and transgenic (Tg) mice.**

A  Prepulse inhibition (PPI) levels (%) of C3H wild-type littermates (*n* = 17) and *Mpst*-KO C3H (*n* = 17) mice. The *X*-axis shows prepulse levels. The pulse sound level was 115 dB.

B  Acoustic startle response (ASR) levels of inbred B6 (*n* = 12) and C3H (*n* = 12) mice. The unit is arbitrary. The *X*-axis shows pulse sound levels.

C  ASR levels of C3H wild-type littermates (*n* = 17) and *Mpst*-KO C3H mice (*n* = 17). The unit is arbitrary. The *X*-axis shows pulse sound levels.

D  PPI levels of B6 wild-type littermates (*n* = 17) and *Mpst*-Tg B6 mice (*n* = 19). The *X*-axis shows prepulse levels. The pulse sound level was 115 dB.

E  ASR levels of B6 wild-type littermates (*n* = 17) and *Mpst*-Tg B6 mice (*n* = 19). The unit is arbitrary. The *X*-axis shows pulse sound levels.

F  $H_2S$ content in the frontal cortex of C3H wild-type littermates (*n* = 7) and *Mpst*-KO C3H mice (*n* = 7).

G  $H_2S$ content in the frontal cortex of B6 wild-type littermates (*n* = 7) and *Mpst*-Tg B6 mice (*n* = 7).

H, I  Levels of acid-labile sulfur (H) and bound sulfane sulfur (I) in the frontal cortex of C3H wild-type littermates (*n* = 7) and *Mpst*-KO C3H mice (*n* = 7).

J, K  Levels of acid-labile sulfur (J) and bound sulfane sulfur (K) in the frontal cortex of B6 wild-type littermates (*n* = 7) and *Mpst*-Tg B6 mice (*n* = 7).

Data information: The values represent the mean ± SEM. *P* values were calculated using Sidak's multiple comparisons test (A–E) or unpaired two-tailed *t*-test (F–K).

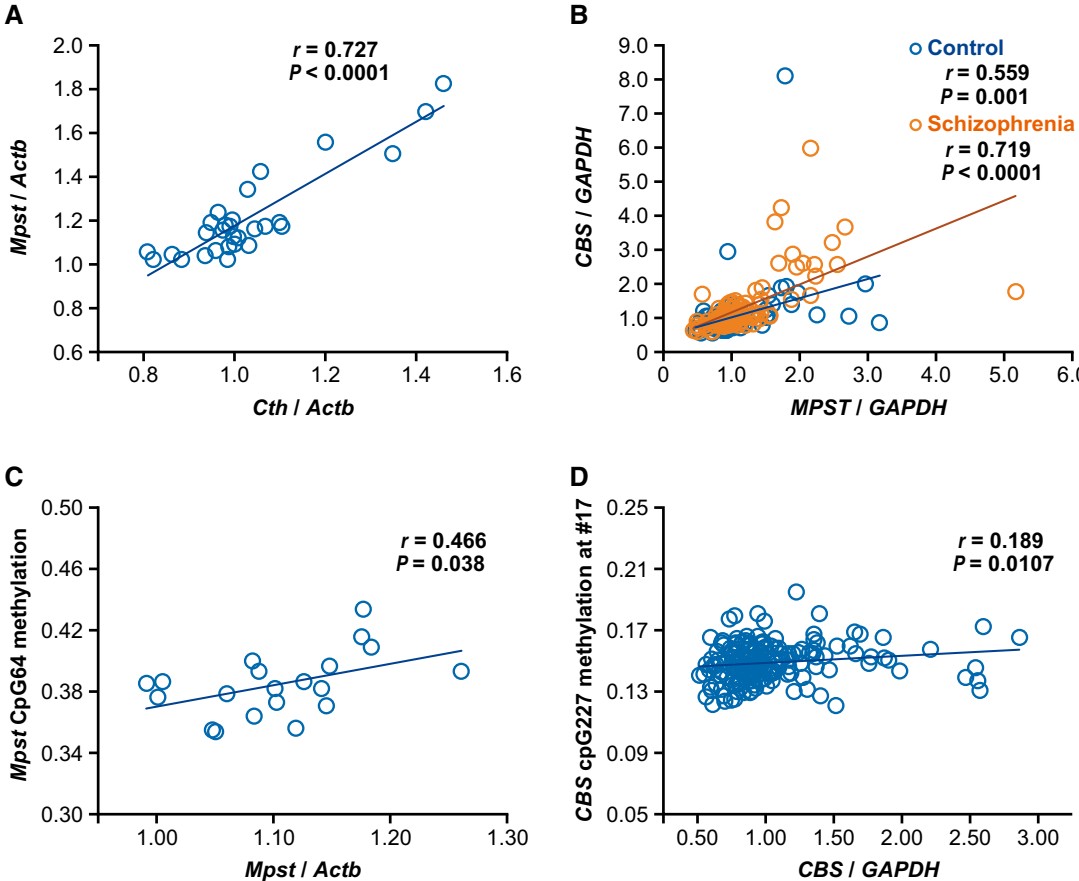

**Figure 5. Correlation between expression levels of genes encoding H₂S-producing enzymes and between DNA methylation levels and gene expression levels.**

A  Correlation between relative expression levels of *Mpst* and *Cth* in B6 (*n* = 15) and C3H (*n* = 14) mice.

B  Correlation between relative expression levels of *MPST* and *CBS* in postmortem brain samples (control, *n* = 93; schizophrenia *n* = 95). For demographic data, see Appendix Table S7.

C  Correlation between DNA methylation levels and relative expression levels of *Mpst* in B6 (*n* = 10) and C3H (*n* = 10) mice. Also see Appendix Fig S14.

D  Correlation between DNA methylation levels and relative expression levels of *CBS* in postmortem brain samples from both control subjects and subjects with schizophrenia (*n* = 181). Also see Appendix Fig S17, and Appendix Table S7 for demographic data.

Data information: Statistical evaluations were performed using Spearman's rank correlation test.

pathophysiology (Lewis *et al*, 2005), and PV-positive GABA neurons are energy-demanding because of high-frequency firing (Tremblay *et al*, 2016). The expression levels of *Pvalb* were lower in the *Mpst*-Tg than in the non-Tg mice in RNA-seq analysis (Fig 8E) and in qRT–PCR analysis (Fig 8F), suggesting that deficits in the bioenergetic pathways could lead to functional impairments in PV-positive interneurons.

Then we analyzed the concentrations of ATP (adenosine triphosphate) and ADP (adenosine diphosphate) and the ATP-to-ADP ratio (Tantama & Yellen, 2014) as energy metabolism indices (Fig 9A–C). In the frontal cortex of *Mpst*-Tg mice, the levels of ATP and ATP-to-ADP ratio were lower than in the non-Tg mice (Fig 9A and C). To confirm the energetic dysregulation of *Mpst*-Tg mice, we analyzed their mitochondrial activity *in vivo*. We focused on cytochrome c oxidase (complex IV), a component of mitochondrial respiratory chain, because excessive H₂S is known to inhibit cytochrome c oxidase in a non-competitive manner (Goubern *et al*, 2007; Cooper & Brown, 2008; Hancock & Whiteman, 2016). The enzymatic activity

of cytochrome c oxidase in mitochondria isolated from the *Mpst*-Tg mice was significantly lower than the non-Tg littermates (Fig 9D).

To further examine the effect of dampened energy metabolism on neural functions, we analyzed the density of dendritic spines using primary culture of hippocampal neurons. Hippocampus is a critical region involved in the neural circuit for the regulation of PPI (Swerdlow *et al*, 2016). The spine density in dissociated hippocampal neurons was found to be diminished in the *Mpst*-Tg mice (Fig 9E and F) compared to that of non-Tg littermates, which is consistent with the result of a human postmortem study (Rosoklija *et al*, 2000).

As an additional possibility, we assessed protein palmitoylation. Specific palmitoyltransferases mediate the transfer of palmitoyl groups to specific cysteine residues on a variety of proteins, modulating the intracellular localization of the target proteins. Intriguingly, Pinner *et al* reported that the levels of protein *S*-palmitoylation in most of the proteins tested were significantly reduced in postmortem brains from schizophrenia patients (Pinner

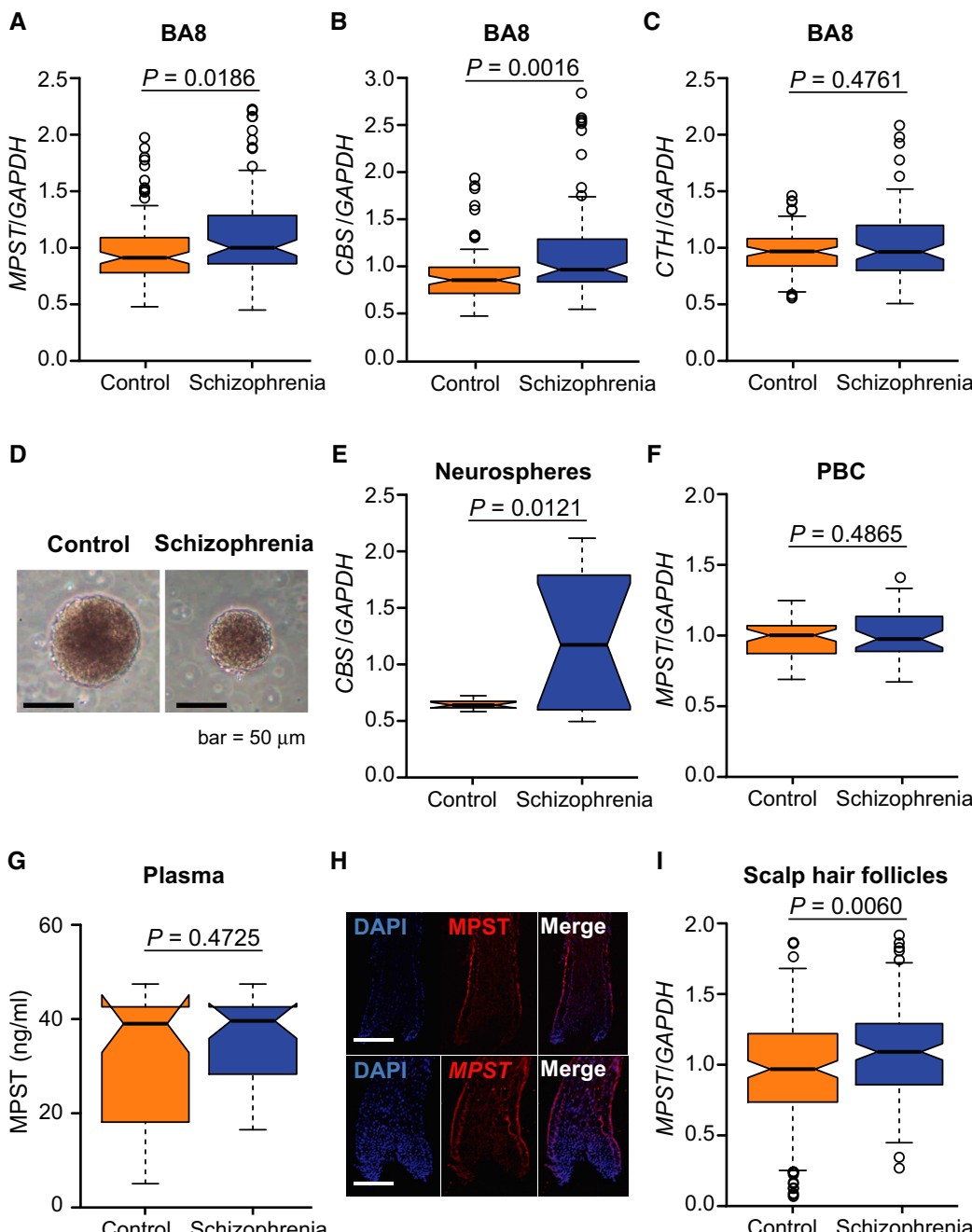

**Figure 6. H₂S-producing genes/proteins in human samples.**

A–C Expression of *MPST* (A), *CBS* (B), and *CTH* (C) in postmortem brain samples from controls (*n* = 93) and schizophrenia patients (*n* = 95) (1st set). For demographic data, see Appendix Table S7.

D Neurospheres from a control (left panel) and a schizophrenia patient (right panel). Scale bar, 50 μm.

E Relative expression of *CBS* in neurosphere samples from controls (*n* = 12 cell lines) and schizophrenia patients (*n* = 12 cell lines).

F Relative expression of *MPST* in peripheral blood cell (PBC) samples from controls (*n* = 56) and schizophrenia patients (*n* = 44). For demographic data, see Appendix Table S9.

G Protein expression levels of MPST in plasma samples from controls (*n* = 56) and schizophrenia patients (*n* = 44). For demographic data, see Appendix Table S9.

H Expression patterns of MPST/*MPST* in scalp hair follicles. The upper panels show MPST protein expression, and the lower panels show *MPST* mRNA expression. Nuclei were stained with DAPI (4′,6-diamidino-2-phenylindole, dihydrochloride). Scale bars, 150 μm.

I Relative expression of *MPST* and *CBS* in scalp hair follicle samples from controls (*n* = 166) and schizophrenia patients (*n* = 149). For demographic data, see Appendix Table S10.

Data information: The boxplot graphs summarize statistical measures (median, the 75th and 25th percentiles, and minimum and maximum data values) of the distributions of relative target gene expression levels (using *GAPDH* as an internal control). BA8, Brodmann area 8 of postmortem brains. *P* values were calculated using two-tailed Mann–Whitney *U* test.

et al, 2016). We tested whether an elevated sulfide stress condition might inhibit the protein S-palmitoylation reaction, because the condition results in enhanced modification of cysteine residues (protein-Cys-SnH) (Kimura et al, 2015, 2017), thereby altering the reactivity of cysteine residues. However, we did not detect any alteration in the levels of protein S-palmitoylation analyzed by means of the modified ABE (acyl-biotin exchange) method (Forrester et al, 2011) in brains from Mpst-Tg or KO mice (Appendix Figs S8–S11).

## DNA methylation levels are correlated with the expression levels of genes for $H_2S$ synthesis

There were no copy number variations for Mpst in the B6 and C3H genomes that can explain the differential expression (Appendix Table 15). By sequencing analysis of the 5′-upstream region of Mpst (~2,600 bp), we detected distinct guanine (G) nucleotide stretches between the B6 ($G_{10}$) and C3H genomes ($G_{12}$) (Appendix Fig S12), which, however, does not seem to explain the differential gene expression. We tested the genetic association of MPST and CBS with schizophrenia by analyzing SNPs in more than 2,000 cases and a comparable number of controls (Bangel et al, 2015; Balan et al, 2017), but detected no association signals (Appendix Table 16). These results suggest that the differential expression of Mpst/MPST and CBS in mouse strains/disease brains is not due to inherent genetic variations. Therefore, we examined the epigenetic mechanism underlying the changes in expression.

We examined the DNA methylation status of the relevant genes. There are two CpG islands at the Mpst locus in mice, one in the promoter region ("CpG28") and the other around exon 2 ("CpG64") (Appendix Fig S13). In "CpG64", the CpG_31, CpG_42.43.44 (examined by probing the interval #1), and CpG_15 (examined by probing the interval #7) sites were highly methylated in the C3H genome compared to the B6 genome (Appendix Fig S14). The mean methylation levels of the three sites were positively correlated with the expression levels of Mpst (Fig 5C).

For the human genes, the MPST locus has two CpG islands, as observed in mice (Appendix Fig S15). There were multiple differentially methylated cytosine nucleotides between schizophrenia and control postmortem brain samples, but methylation levels at those sites did not show any significant correlation with the expression levels of MPST. In the CBS genomic region, there is one CpG island in the intron 1 interval (Appendix Fig S16). In the postmortem brain samples, the average methylation levels in the 3′ portion of "CpG227" (examined by probing the interval #17) were higher in the schizophrenia samples than in the controls (Appendix Fig S17) and exhibited a positive correlation with the expression levels of CBS (Fig 5D). In neurosphere samples, the 3′ portion of "CpG227" (examined by probing the interval #17) from schizophrenia with 22q11.2 deletion also showed higher methylation levels than that from controls (Appendix Fig S18).

These results suggest that least elevated expression levels of Mpst in C3H brains and CBS in the schizophrenia postmortem brain samples and iPSCs-derived neurospheres could be elicited by altered DNA methylation levels in the CpG islands in the target gene regions.

## Inflammatory/oxidative stress in the early brain developmental stage results in upregulated $H_2S$/polysulfides production

$H_2S$ plays a protective role against oxidative and inflammatory stresses (Kwak et al, 2003; Niu et al, 2015). Oxidative stress depletes GSH and cysteine, which in turn enhances the activity of $H_2S$/polysulfides-producing enzymes to stimulate the synthesis of $H_2S$, cysteine, and GSH, leading to the protection of cells from oxidative stress (Appendix Fig S19) (Kimura, 2010; Niu et al, 2015). $H_2S$ also attenuates the phosphorylation of mitogen-activated protein kinases as well as the activation of NF-κB and caspase-3 and suppresses Bcl-2 expression under inflammatory conditions (Tripatara et al, 2008). Based on the above, when oxidative insults occur in an early neurodevelopmental stage, the expression of both typical antioxidant and $H_2S$/polysulfides synthesis genes could be programmed to be maintained at higher levels via epigenetic mechanisms. To examine this hypothesis, we used a mouse model of maternal polyriboinosinic-polyribocytidilic acid (poly-I:C) injection (maternal immune activation model) (Meyer & Feldon, 2012; Giovanoli et al, 2013; Bundo et al, 2014), which is known to perturb early neural development via inflammatory/oxidative insults. Pregnant mice (B6) received five intraperitoneal injections of poly-I:C during embryonic days 12–16 (E12–E16), and gene expression in the cortex was analyzed at 13–14 weeks of age (Fig 10A). In this protocol, a "schizophrenia endophenotype of increased L1 copy number" was observed (Bundo et al, 2014). Interestingly, the expression levels of Mpst and Cbs were elevated in the adult brain of this model, albeit Cth expression was unchanged (Fig 10B–D).

As representative antioxidant genes, we examined the expression of Sod1 and Sod2 (encoding superoxide dismutase), Cat (encoding catalase), and Gpx1 and Gpx4 (encoding glutathione peroxidase) (Smaga et al, 2015). The expression levels of all these genes were significantly upregulated in the poly-I:C administered mice compared to the controls (Fig 10E–I), although the fold changes were small (for poly-I:C-to-control, Sod1 = 1.043, Sod2 = 1.024, Cat = 1.051, Gpx1 = 1.050, Gpx4 = 1.079), suggesting small effect sizes of individual expressional changes. The relative expression levels of the two genes encoding $H_2S$/polysulfides synthesis were compared in the following pairwise comparisons: Mpst versus Cbs (Fig EV4A), Mpst versus individual antioxidant genes (Fig EV4B–D), and Cbs versus individual antioxidant genes (Fig EV4E–H). Except for Mpst versus Gpx1, all the pairwise comparisons exhibited positive correlation.

Conforming to the above "inflammatory/oxidative stress in the early brain developmental stage" theory, inflammatory genes Il1b, Il6, and Tnf were upregulated in C3H brain compared to B6 at some point(s) during E (embryonic day) 16. 5 to P (postnatal day) 7 without downregulation in any periods, though downregulation of Il6 was seen at an adult stage (12 weeks old) (Fig 10J–L). These developmental events could lead to an elevated $H_2S$/polysulfides production state in C3H mice compared to B6.

In human brains, CAT was upregulated in schizophrenia samples compared to the controls (Fig EV5A), and the expression level of this gene was positively correlated with that of MPST and CBS in both the schizophrenia and control samples (Fig EV5B and C). The IL1B/Il1b, IL6/Il6, or TNF/Tnf was not significantly upregulated in the schizophrenia postmortem brain sample (Toyoshima et al,

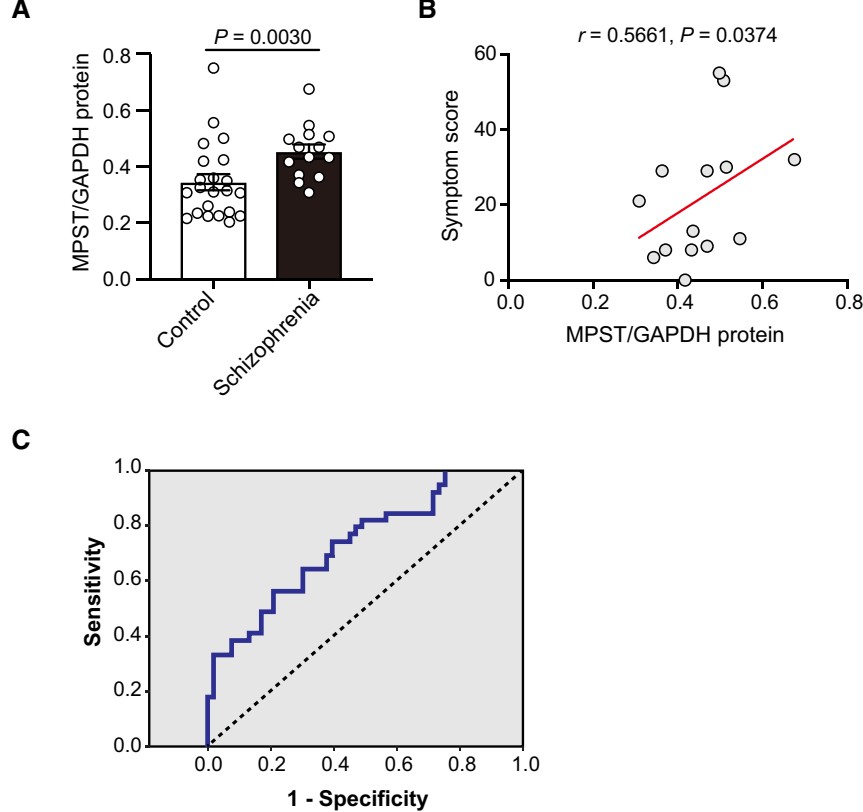

**Figure 7. Assessments of MPST in postmortem brains, and its association with symptom severity score and *MPST* in hair follicle samples as a biomarker.**

A   Protein expression levels of MPST in postmortem brain samples from controls ($n$ = 22) and schizophrenia patients ($n$ = 14) (2$^{nd}$ set). Also see Appendix Table S8 for demographic data.

B   Correlation between relative expression levels of MPST in postmortem brain samples from schizophrenia ($n$ = 14) and symptom severity score rated by using the Diagnostic Instrument for Brain Studies (DIBS). Also see Appendix Table S8 for demographic data.

C   Receiver Operating Characteristic (ROC) curve of *MPST/GAPDH* levels in the hair follicle sample set, relative to schizophrenia occurrence. Schizophrenia patients ($n$ = 149) are the same to those used in Fig 6I. Regarding the discrimination efficiency of schizophrenia, the sensitivity and specificity were 73.6 and 47.2%, respectively (area under the curve = 0.602).

Data information: The values represent the mean ± SEM (A). *P* value was calculated using two-tailed Mann–Whitney *U* test (A). Correlation analysis was performed using Spearman's rank correlation test (B).
Source data are available online for this figure.

2016) and in the poly-I:C model mice (Appendix Fig S20). These results support our hypothesis that upregulated H$_2$S/polysulfides production in adulthood can stem from oxidative/inflammatory insults in the neurodevelopmental stage, occurring in a concerted manner with the elevation of antioxidant gene expression and via an epigenetic mechanism.

## Discussion

Prior studies have repeatedly reported physiologically necessary/ beneficial and neuroprotective roles of H$_2$S/polysulfides as gaso-transmitters or neuromodulators (Szabo, 2007; Kimura, 2015; Wallace & Wang, 2015), including those in Huntington's disease

**Figure 8. RNA-seq and RT–PCR analyses of *Mpst*-KO and Tg mice.**

A, B   Volcano plot of RNA-seq analysis shows differentially expressed genes in the frontal cortex between *Mpst*-KO mice (A) and *Mpst*-Tg mice (B) in comparison to their wild-type littermates; $n$ = 6 in each group. Green and blue dashed lines indicate *P* value thresholds of 0.05 and 0.01, respectively. Top hits among the differentially expressed genes (*P* < 0.05 and absolute fold change > 2) are highlighted as red in the plot.

C   Gene ontology enrichment analysis (biological process) of dysregulated genes in *Mpst*-Tg mice visualized using a volcano plot for the enriched terms.

D   Differentially expressed genes in *Mpst*-KO mice (upper panels) and *Mpst*-Tg mice (lower panels) (*P* < 0.05) in comparison to their WT or non-Tg littermates were analyzed for the enriched canonical pathways.

E, F   Gene expression of *Pvalb* in non-Tg and *Mpst*-Tg mice examined by RNA-seq ($n$ = 6 in each group) (E) and by qRT–PCR analysis ($n$ = 10 in each group, but an outlier was excluded in WT) (F).

Data information: The values represent the mean ± SEM. *P* values were calculated using unpaired two-tailed *t*-test (E, F).

▶

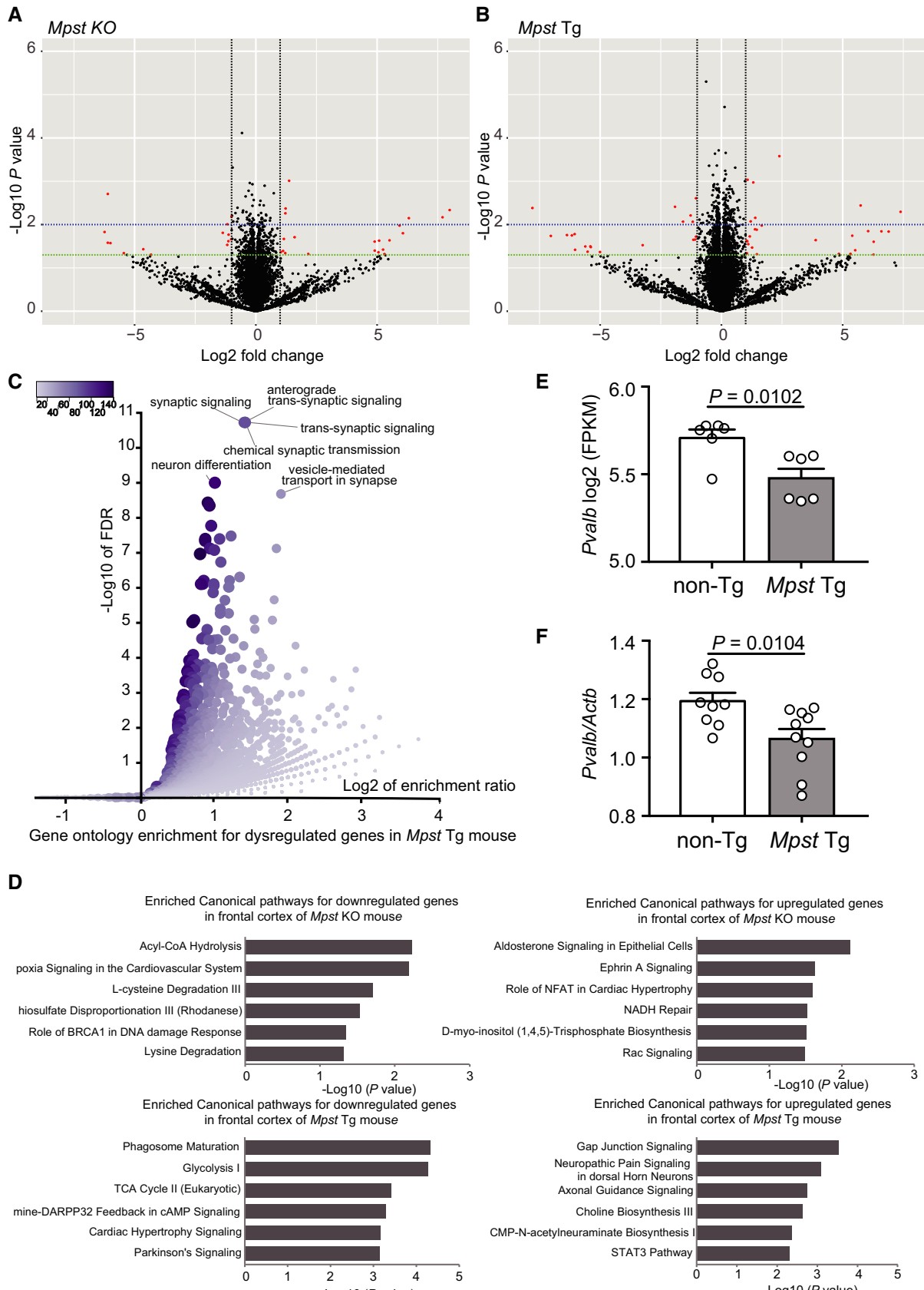

**Figure 8.**

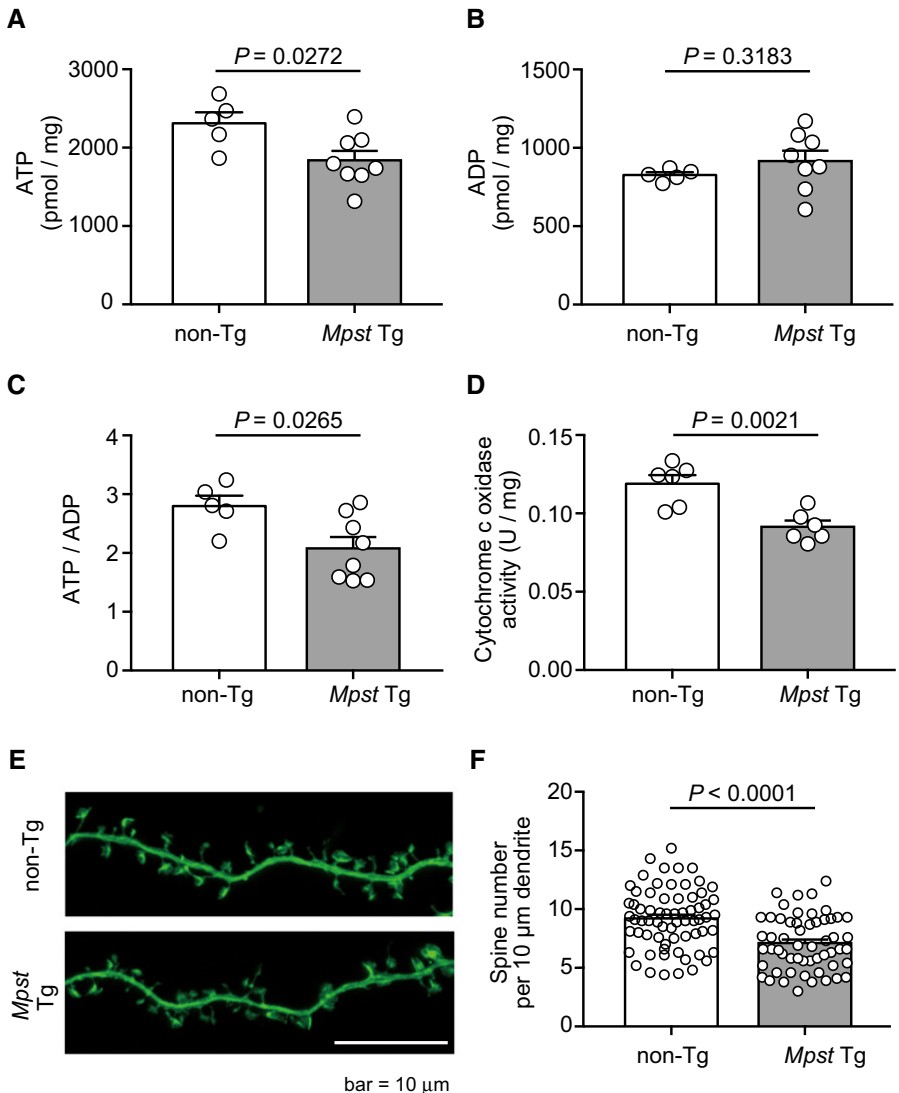

**Figure 9. Energy metabolism and spine analyses of *Mpst*-Tg mice.**

A–C   Concentrations of ATP and ADP, and ATP-to-ADP ratios in non-Tg (n = 5) and *Mpst*-Tg (n = 8) mice (both age of 6 weeks).

D      Cytochrome c oxidase activity in non-Tg (n = 6) and Tg (n = 6) mice (both age of 6 weeks).

E, F   Fluorescence images of dendritic spines of dissociated hippocampal neurons (E16.5) of non-Tg and *Mpst*-Tg mice (E) and their quantification (F). N = 50–70 images from 10 to 15 cells. Scale bar, 10 μm.

Data information: The values represent the mean ± SEM. P values were calculated using unpaired two-tailed *t*-test.

(Paul *et al*, 2014). In contrast, in this study, we started by analyzing the molecular underpinnings of the sensorimotor gating function against acoustic stimuli in mice and, as a consequence, provide the first evidence that excess $H_2S$/polysulfides production could underlie the pathophysiology of, at least, a subset of schizophrenia. Although we revealed the elevated sulfide stress in schizophrenia pathophysiology, in the light of Mpst from our proteomics study, upregulation of a combination of the three genes encoding $H_2S$/polysulfides-synthesizing enzymes, namely, *Mpst* (*MPST*), *Cbs* (*CBS*), and *Cth* (*CTH*), was observed in mouse models and human schizophrenia samples. Regarding such combination, we speculate that depending on the tissue-specific variability in the expression levels of these genes, a gene network including predominantly

expressed one in the relevant tissue (see Fig EV3) may leverage maximum control of $H_2S$/polysulfides production. Hydrogen sulfide is a highly bioactive agent in organisms, thus even small changes in its production levels should have serious consequences in life. Therefore, we believe that even small changes detected in this study can be biologically meaningful. We further demonstrated that altered expression of these genes could be, at least in part, primed by changes in genomic DNA methylation markers, although their direct link remains to be proved. We demonstrated that such epigenetic changes can be traced back to inflammatory/oxidative insults in the early brain developmental stage by analyzing the poly-I:C model, which is consistent with, and a novel aspect of, the "neurodevelopmental theory" of schizophrenia pathogenesis. We

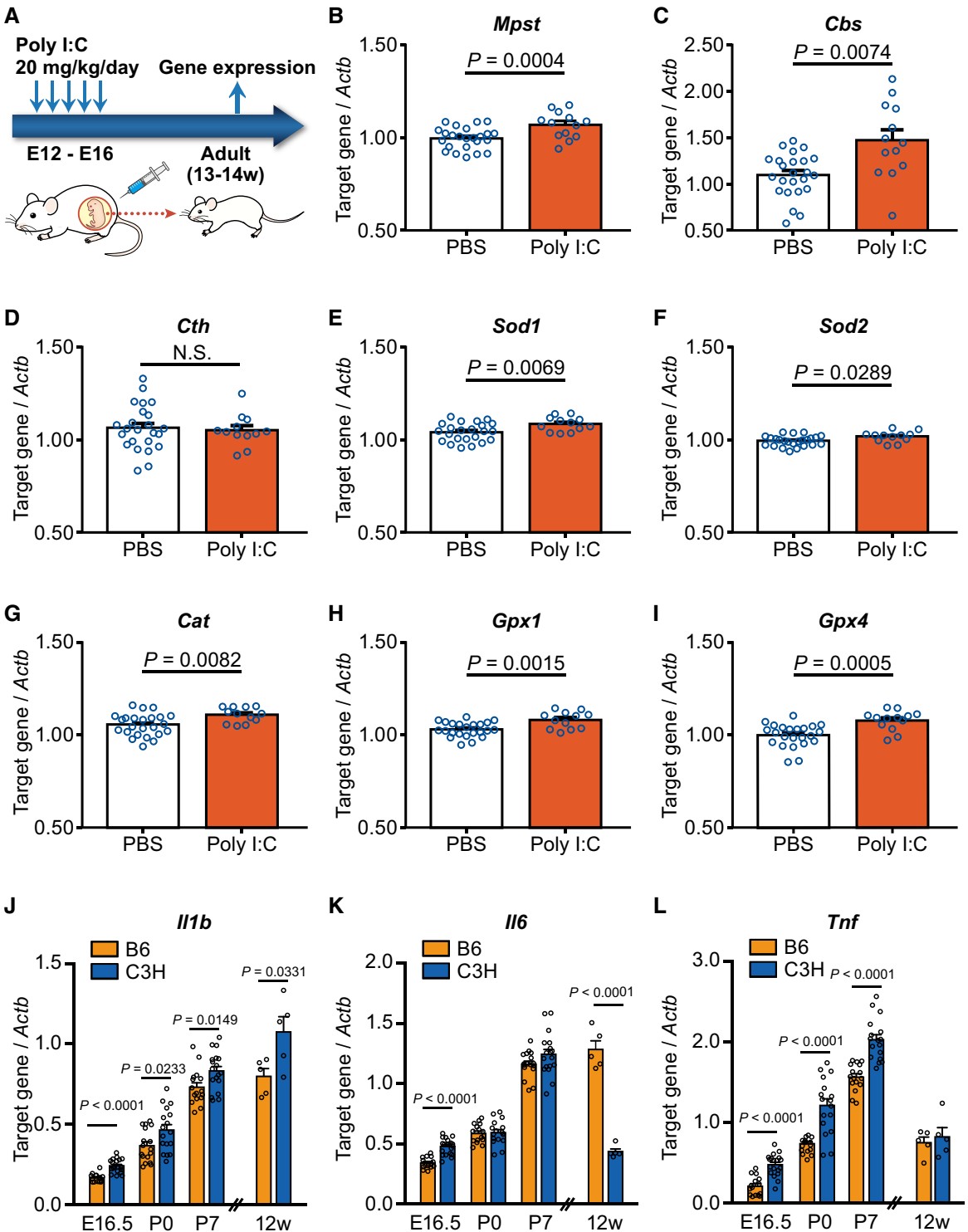

**Figure 10. Gene expression in the maternal immune activation mouse model and in multiple developmental stages.**

A   Schematic illustration of the experimental design of maternal immune activation mouse. B6 mice were used, and gene expression was examined in the frontal cortex.

B–I   Target gene expression levels were relative to that of *Actb*. PBS-injected mice, *n* = 25; poly-I:C-injected mice, *n* = 13.

J–L   Comparison of expression levels of typical inflammatory genes *Il1b* (J), *Il6* (K), and *Tnf* (L) in four developmental points of brains between B6 and C3H mice. "12W" samples were analyzed separately from the others. Whole cortex for E16.5 (*n* = 15–20), P0 (*n* = 17), and P7 (*n* = 17–18), and frontal cortex for 12W (*n* = 5) were used for analysis. E16.5, embryonic day 16.5; P0, at birth; P7, postnatal day 7; 12W, 12 weeks of age.

Data information: The values represent the mean ± SEM. *P* values were calculated using unpaired two-tailed *t*-test.

reason that upregulation of the $H_2S$/polysulfides synthesis genes and representative antioxidant genes can be orchestrated against an oxidative event in a complementary manner because $H_2S$/polysulfides can mitigate oxidative stress (Kimura, 2010; Niu et al, 2015; Wallace & Wang, 2015). Notably, endogenous $H_2S$ also enhances the activities of catalase and superoxide dismutase in bacteria (Shatalin et al, 2011).

To determine the functional consequence of excess $H_2S$/polysulfides production under disease conditions, we performed RNA-seq analysis using Mpst-KO and Tg mice and detected the DARPP32 and energy metabolism (glycolysis and TCA cycle) systems among the top canonical pathways, dysregulation of which has been documented in schizophrenia (Kunii et al, 2014; Wang et al, 2017; Zuccoli et al, 2017). In particular, several lines of investigation have suggested a reduced capacity of the TCA cycle in schizophrenia (Bubber et al, 2011; Paredes et al, 2014). Concordant with those reports, our Mpst-Tg mice showed reduced ATP concentration and decreased ATP-to-ADP ratio, and deficits in cytochrome c oxidase (complex IV) activity, compared to the non-Tg animals. These dampened bioenergetic processes may have led to decreased Pvalb expression, a functional marker of PV-positive interneurons, and diminished spine density. The brain is deemed to be susceptible to $H_2S$, because in the brain, the activity of SQOR (sulfide quinone oxidoreductase), a $H_2S$-metabolising enzyme, was lower than in peripheral tissues (Linden et al, 2012). The detailed mechanistic links between these issues, including in vivo stoichiometry of $H_2S$/$H_2S_n$ on the suppression of complex IV activity (Szabo et al, 2014; Modis et al, 2016), should be clarified in future.

There are several cases in which excess $H_2S$ production results in cognitive impairments. The CBS gene is encoded on chromosome 21 (21q22.3), which is associated with trisomy in Down syndrome (DS). Overexpression of CBS may cause developmental abnormalities in cognition in children with DS (Ichinohe et al, 2005). Barbaux et al (2000) reported that the CBS 844ins68 allele was significantly underrepresented in children with high IQs. The CBS 844ins68 allele carries a 68-bp insertion consisting of a duplicated intron 7/exon 8 boundary (Sebastio et al, 1995) and exhibits enhanced activity compared to the WT allele (Barbaux et al, 2000). Recent evidence has showed that increased copy number of Cbs elicits memory-related cognitive deficits in DS mouse models, by perturbing synaptic activity, possibly through $H_2S$ signaling (Marechal et al, 2019). These results suggest that excess $H_2S$ induced by enhanced CBS/Cbs activity may contribute to at least some aspect(s) of cognitive impairments manifested in schizophrenia (Rajagopal et al, 2014). Similarly, ethylmalonic encephalopathy (EE; OMIM #602473), an autosomal recessive mitochondrial disease associated with progressive neurological failure, is caused by mutations in ETHE1 (Burlina et al, 1991; Mineri et al, 2008). ETHE1 encodes a ubiquitous mitochondrial sulfur dioxygenase (SDO) (Tiranti et al, 2009) that eliminates $H_2S$ (Hildebrandt & Grieshaber, 2008).

Because $H_2S$ is distributed systemically, some somatic diseases are also affected by $H_2S$. Ulcerative colitis is an inflammatory bowel disease, and $H_2S$ overproduction contributes to the etiology of this disease (Roediger et al, 1997). On the other hand, $H_2S$ is present in joints and acts as a proinflammatory mediator (Muniraj et al, 2017); therefore, rheumatic diseases might be associated with decreased production of $H_2S$. Comorbidity has been reported between

ulcerative colitis and schizophrenia (Cucino & Sonnenberg, 2001), whereas rheumatoid arthritis occurs at a relatively low frequency in schizophrenia patients (Leucht et al, 2007).

In schizophrenia, biomarkers are of cardinal importance for early intervention and improved prognoses (Morrison et al, 2012; Fusar-Poli et al, 2013). Biomarkers can be also used to categorize patients and determine optimal therapeutics. Our postmortem study suggested a positive correlation between symptomatic severity and "sulfide stress", and in living patients the present results highlight the potential benefit of examining MPST expression levels in scalp hair follicles. The use of scalp hair follicles is advantageous because of the convenience and noninvasiveness of sampling (Maekawa et al, 2015).

Meanwhile, enhanced inflammatory conditions in schizophrenia are repeatedly reported, in particular around onset and in active phase, which may be state-dependent (temporal) (Smaga et al, 2015; Koga et al, 2016; Watkins & Andrews, 2016). Excess $H_2S$/polysulfides production may be a more basic and persistent phenomenon like, "continuous bass", in the realm of "reductive response" which has not been explicitly addressed in the previous studies. The precise contribution of these two biological reactions to disease process remains elusive.

In summary, although $H_2S$/polysulfides are necessary for maintenance of normal cellular functions, an excessive amount of $H_2S$/polysulfides may impair brain functions; we propose to name this phenomenon "sulfide stress", which could be a novel form of "reductive stress" (Perez-Torres et al, 2017). One of the potential origins of "sulfide stress" could be oxidative/inflammatory insults in neurodevelopment and is epigenetic priming as a mechanism, and its consequence could include damped energy metabolism. Elucidation of the precise role of "sulfide stress" in schizophrenia and other mental disorders will facilitate the establishment of a novel paradigm for drug development.

# Materials and Methods

### Experimental animals

For the proteomics study, B6 (C57BL6/NCrj) and C3H (C3H/HeNCrj) mice were purchased from Japan SLC (Shizuoka, Japan). For the other studies, B6 (C57BL/6NCrl) and C3H (C3H/HeNCrl) mouse strains were obtained from Japan's Charles River Laboratories (Kanagawa, Japan). The change in substrains was due to the introduction of the International Genetic Standardization Program (https://www.crj.co.jp/cms/cmsrs/pdf/company/rm_rm_r_igs.pdf) after the proteomics study. The animals were bred in our facilities for 2 weeks before the experiments, housed by strain in groups of four, using standard cages in a temperature- and humidity-controlled room with a 12-h light/dark cycle (lights on at 0800). The animals had free access to standard laboratory chow and tap water. The experimental procedures conformed to NIH guidelines for animal welfare and were approved by the Animal Ethics Committees of all the relevant institutes. The mouse studies contained no randomization procedures for group allocation and were conducted in an open-label format. Only male animals were used for all the experiments except for the data for Fig 10J–L, samples for E16.5, P0, and P7.

## Sample preparation for proteomics analysis

Eight-week-old mice were decapitated, and the bilateral prefrontal cortex (AP > +2 mm from the bregma) of each brain was dissected and frozen immediately. The spleen was also dissected, rinsed in ice-cold phosphate-buffered saline, homogenized on a stainless steel fine mesh net, and filtered using Eagle's MEM supplemented with 10% FBS. The filtered cells were centrifuged, and the supernatants were removed. Red blood cells were ruptured in an $NH_4Cl$ solution and washed with Eagle's MEM, and then, residual lymphocytes were collected. All lymphocyte and brain samples were stored at −80°C until protein preparation. Tissues were homogenized and sonicated in ice-cold lysis buffer (20 mM Tris, 7 M urea, 2 M thiourea, 4% (w/v) CHAPS, 10 mM DTT, 1 mM EDTA) containing protease and a phosphatase inhibitor cocktail (P8340, P2850, P5726; Sigma, St Louis, MO, USA). After centrifugation of the lysate ($1.7 \times 104$ $g$), the supernatant was recovered and assayed for protein concentration using the Bradford method.

## 2D-DIGE analysis of brain and splenic lymphocyte samples from mice: labeling

The experimental design for the 2D-DIGE analysis in this study is shown in Appendix Table S1. Mixtures from each sample (100 mg of protein) derived from brain or lymphocytic tissue were pooled as an internal standard. A 100-μg aliquot of protein from each sample was labeled with Cy3 for B6 mice, Cy5 for C3H mice, and Cy2 for the internal standard and left on ice for 30 min in the dark. The labeling reaction was quenched with 2 μl of 10 mM lysine. Each pair of labeled B6- and C3H-derived samples was then combined with the labeled internal standard and added to an equal volume of 2× sample buffer [9 M urea, 4% (w/v) CHAPS, 2.4% (v/v) DeStreak reagent (GE Healthcare, Chicago, IL, USA), 1% (v/v) IPG buffer 3–10 (GE Healthcare)] and diluted with rehydration buffer [9 M urea, 2% CHAPS, 1.2% (v/v) DeStreak reagent, 0.5% (v/v) IPG buffer 3–10], to a final volume of 250 μl, prior to loading onto the 2D gel.

## 2D gel electrophoresis

One Immobiline DryStrip [pH 3–10, 13 cm (GE Healthcare)] was passively rehydrated with each combined sample for more than 10 h at room temperature. For the first dimension of electrophoresis, the rehydrated strips were focused using Ettan IPGphor II (GE Healthcare) for a total of 82,000 Vh. The strips were then reduced and alkylated according to the manufacturer's protocol. For the second dimension of gel electrophoresis, the strips were loaded onto 12% polyacrylamide gels in a Hoefer SE 600 Chroma system, and proteins were resolved at 15 mA/gel for 7 min and then at 30 mA/gel for the duration of the run.

## Data analysis for 2D gel electrophoresis

Separated gels were scanned using a Typhoon 9400 digital scanner (GE Healthcare), and images were analyzed by differential analysis software (DeCyder, version 5.0, GE Healthcare). Spots were recognized using a Differential In-gel Analysis (DIA) module, with the estimated number of spots for each co-detection procedure set to 5,000 and filtered using default settings. Intergel matching and statistical analyses were performed using the Biological Variable Analysis (BVA) module.

## Protein identification: SYPRO ruby and silver staining

Three-hundred-microgram aliquots of protein from each tissue sample were diluted in rehydration buffer to a final volume of 250 μl and separated by 2D gel electrophoresis as described above. The gels were then stained with SYPRO Ruby Protein Gel Stain (Molecular Probes) according to the manufacturer's protocol. The resulting fluorescent images were scanned. Protein spots were excised and destained with 25 mM $(NH_4)HCO_3$/50% (v/v) acetonitrile (ACN), followed by an additional wash in 100% ACN using Xcise (Shimadzu), and then dried to enable in-gel digestion.

For large-scale analysis, 500 μg of protein was focused using a 24-cm Immobiline DryStrip pH 3–7 (GE Healthcare) for a total of 160,000 Vh. In the second dimension, the run was performed using an Ettan DALT Six (GE Healthcare). Gel spots were stained with a Plus One Silver Stain Kit according to the manufacturer's instructions and excised with an Ettan Spot Picker (GE Healthcare). Gels were then destained in buffer [15 mM potassium hexacyanoferrate (III), 50 mM sodium thiosulfate], washed in 100% ACN, and dried.

## In-gel digestion and MALDI-TOF mass spectrometry (MS)

In-gel digestion was performed using 3 μg/ml trypsin in 50 mM $(NH_4)HCO_3$ with 0.1% (w/v) N-octylglucoside at 37°C for 16 h. Digested peptides were extracted with 5% (v/v) formic acid/50% ACN, dried, diluted in 0.1% (v/v) trifluoroacetic acid (TFA), and then desalted and concentrated in a PerfectPure C-18 tip (Eppendorf). Peptides were eluted onto a MALDI target plate with matrix solution (5 mg/ml α-cyano-4-hydroxycinnamic acid (CHCA), 0.1% TFA, 50% ACN, 500 pmol of bradykinin, 400 pmol of ACTH) and dried at room temperature. The peptide masses were then determined by MALDI-TOF MS. Mass spectra were obtained using an AXIMA CFR plus (Shimadzu, Kyoto, Japan) in positive ion reflectron mode, and the mass axis was calibrated using bradykinin and ACTH. Raw data were processed using Kompact (ver. 2.4, Shimadzu), and the peak list was generated with default parameters as follows: minimum mass: 600, maximum mass: 3,500, minimum isotopes: 2, maximum intensity variation: 50, overlapping distributions: on, and minimum peak percent: 30. The identified proteins were verified by manual comparisons of the computer-generated fragment ion series of the predicted peptide with experimental MS data. The mass spectrum for each spot was analyzed by PMF using MASCOT™ software (ver. 2.2.1, Matrix Science) (http://www.matrixscience.com) and used for PMF analysis against the UniProt or NCBI nonredundant protein databases. Database searching was performed using the following settings: (i) *Mus musculus*, (ii) trypsin digestion with up to 1 missed cleavage, (iii) ± 0.5 Da error, (iv) cysteine carbamidomethylation as a fixed modification, and methionine oxidation as a variable modification. After excluding unmatched peptides to avoid contaminations of overlapping spots, identities were assigned when score values were returned with $P$ value < 0.05 for a false positive and were confirmed by manual inspection of the spectra. Each protein was confirmed with 2D Western blot analyses described below.

## 1D and 2D Western blot analyses: antibodies

We generated rabbit polyclonal antibodies against the N-terminal (DASWYLPKLGRDARREF) and C-terminal (YMRAQPEHIISEGRGKT) regions of mouse Mpst [designated as anti-Mpst #1 and anti-Mpst #2, respectively]. For 1D Western blot analyses, the anti-Mpst #1 and #2 and anti-α-tubulin (T5168, Sigma) antibodies were diluted in 1:100, 1:1,000, and 1:20,000, respectively. The anti-Mpst #1 (dilution 1:500) and commercially sourced antibodies, which included anti-GRP 75 (Sigma; dilution 1:2,000) for Hspa9 and anti-NPM1 (ProteinTech Group, Rosemont, IL, USA; dilution 1:1,000) for Npm1, were used in 2D Western blot analyses of both frontal cortex and lymphocytes. Anti-peroxiredoxin-6 (Abcam, Cambridge, England) for Prdx6 was diluted in 1:1,000 (frontal cortex) and 1:500 (lymphocytes), and anti-NME2 (Abgent, San Diego, CA, USA) for Nme2 was diluted in 1:100 (frontal cortex) and 1:50 (lymphocyte) (expression levels of Prdx6 and Nme2 in lymphocytes were low).

## Western blotting

For 2D Western blotting, 300 μg of protein was separated by 2D gel electrophoresis using a 13-cm Immobiline DryStrip pH 3–10 or pH 4–7 (GE Healthcare) using the same procedure as described above, except for the rehydration buffer [9 M urea, 2% CHAPS, 0.28% (w/v) DTT, 0.5% (v/v) IPG buffer 4–7] in the pH 4–7 strip. Proteins were transferred to PVDF membranes by semidry blotting, and the membranes were stained with Deep Purple (GE Healthcare) according to the manufacturer's protocol. Fluorescent signals were scanned with Typhoon 9400, after which the membranes were soaked in transfer buffer prior to antibody binding. Membranes were incubated in blocking solution containing 5% skimmed milk in TBS with 0.05% Tween 20 (TBS-T) for 1 h and then incubated overnight at 4°C with primary antibodies in blocking solution. The membranes were then washed in TBS-T and probed with horseradish peroxidase (HRP)-conjugated anti-rabbit or anti-mouse IgG antibodies (Sigma). Signals were detected using Western Blotting Luminol Reagent (Santa Cruz Biotechnology, Dallas, Texas, USA). The fluorescent signals on the 2D Western blots were detected using a Typhoon 9400. Chemiluminescent signals were visualized using a Typhoon 9400 or LAS-3000 (Fuji Film, Tokyo, Japan). Fluorescent and chemiluminescent images were merged using FluorSep software (version 2.2, GE Healthcare). For standard 1D Western blotting, 20-μg protein samples were loaded onto 12% SDS–polyacrylamide gels and separated by electrophoresis. Proteins were transferred to PVDF membranes by semidry blotting, and the membranes were incubated overnight at 4°C with primary antibodies in blocking solution. The membranes were then washed in TBS-T and probed with horseradish peroxidase (HRP)-conjugated anti-rabbit (Sigma, cat no. 12-348) or anti-mouse IgG antibodies (Sigma, cat no. 12-349).

## DNA sequencing and database searching

Genomic DNA samples were extracted using a standard phenol/chloroform method from mouse tails and amplified using fluorescently labeled forward and reverse primers and ExTaq polymerase (TaKaRa Bio, Shiga, Japan). Sequencing was performed using the BigDye Terminator Cycle Sequencing FS Ready Reaction Kit ver. 3.1 (Applied Biosystems, Foster City, CA, USA) and an ABI PRISM 3730 genetic analyzer (Applied Biosystems). Genomic sequences and SNPs were examined using the UCSC Genome Bioinformatics online database (http://genome.ucsc.edu/cgi-bin/hgGateway: version Dec. 2011).

## Measurement of $H_2S$, acid-labile and bound sulfur

After 10- to 12-week-old mice were decapitated, brains were quickly removed and sagittally divided into two groups. Using one hemisphere, $H_2S$ levels were measured according to a previously reported method (Kimura et al, 2015). Using the other hemisphere, acid-labile and bound sulfur levels were measured according to methods described elsewhere (Ishigami et al, 2009; Shibuya et al, 2009).

## Generation of *Mpst* knockout (KO) mice

We generated *Mpst*-deficient mice by genome editing using the CRISPR/Cas9 nickase in the genetic background of the C3H strain (Appendix Fig S6). First, the sgRNAs [*Mpst*-upstream (target sequence: 5′-GGAGAGACAAGCCTGCATTCAGG-3′) at intron 1 and *Mpst*-downstream (target sequence: 5′-GTGGGTGGCGGAGGCTCT GAAGG-3′) at exon 2, where the underlined bases indicate the PAM sequences] were synthesized *in vitro* (T7 gRNA Smart Nuclease Synthesis Kit, System Biosciences, Mountain View, CA, USA) following the manufacturer's instructions. C3H zygotes were microinjected with the cocktail [5 ng/ml Cas9 nickase mRNA (System Biosciences) and 5 ng/ml each of the two sgRNAs]. The injected zygotes were transplanted into the uteri of pseudopregnant dams, and the targeted region of the *Mpst* gene from the resultant pups, which were obtained by cesarean section, was examined by direct sequencing of PCR products amplified from the template DNAs extracted from the tail with primer set A (forward: 5′-CTGTTTTGCAGCCCCGCAATCT-3′, reverse: 5′-TAGGTACCAGGACG CGTCCAGTAA-3′). Mutated alleles of the promising founders were further analyzed by sequencing of the PCR products that were subcloned into pCR2.0 (Invitrogen, Grand Island, NY, USA). When the selected founder (#461) carrying a 4-bp deletion downstream of the start codon in exon 2 of *Mpst* reached sexual maturity, *in vitro* fertilization was performed with C3H strain-derived oocytes to obtain heterozygous mice harboring the mutated allele. The heterozygous males and females, at sexual maturity, were intercrossed to produce homozygotes and control littermates. Routine genotyping of the mice was conducted on an ABI 3130xI genetic analyzer (Life Technologies, Carlsbad, CA, USA) followed by genomic PCR using primer set A, which produced 152- and 156-bp fragments from the deleted allele and unedited allele, respectively. To confirm loss of the Mpst protein in homozygotes, the brain tissues of the animals were subjected to Western blot analysis with an anti-Mpst antibody (Nagahara et al, 1998).

## Generation of *Mpst*-transgenic (Tg) mice

*Mpst*-Tg mice were generated as described elsewhere (Ohnishi et al, 2010) with minor modifications, using the pCAGGS backbone harboring the CAG promoter (Niwa et al, 1991) (Appendix Fig S7) in the genetic background of B6. The HA-tagged *Mpst* ORF derived from B6 was cloned into the *Xho*I site of pCAGGS, and the resultant plasmid was microinjected into B6 zygotes after linearization with *Sca* I. The injected zygotes were transplanted into the uteri of

pseudopregnant ICR dams. As a primary screening, genomic DNAs from the tail of the pups were subjected to genomic PCR to identify animals with the transgene using the three primer pairs covering different parts of the construct (pair A: *pCAGGS_FW2*: 5′-TAACCATGT TCATGCCTTCT-3′ and *Mpst_Rv1*: 5′-TTTACTGAGCCAGGGATGT-3′, pair B: *Mpst_Fw1*: 5′-ATCTACGACGACAGTGACCA-3′ and *β-globinA_Rv1*: 5′-GTCGAGGGATCTCCATAAGA-3′, and pair C: *CAG_Fw4*: 5′-ACCTGGGTCGACATTGATTA-3′ and *CAG_Rv2*: 5′-AACATGGT TAGCAGAGGCTC-3′). As a secondary screening, proteins extracted from the tail were subjected to SDS–PAGE and Western blotting with anti-*Mpst* and anti-HA antibodies to identify HA-Mpst-positive animals. Here, notably, the CAG promoter is known to be ubiquitously active among various tissues, including the tail and brain, enabling us to expect high-level expression in the brains of animals that exhibited transgene expression in the tail. Since founder #737 showed the highest expression of the transgene among the candidates, the offspring animals of this founder were generated. Transgene expression was examined by Western blot analysis.

To identify the integration site of the transgene, next-generation sequencing of genomic DNA obtained from the tail was performed by TaKaRa Bio. The raw sequence data were analyzed using multiple tools, including BWA (http://biobwa.sourceforge.net/), Picard (http://picard.sourceforge.net/), BEDTools (http://code.google. com/p/bedtools/), BreakDancer (http://breakdancer.sourceforge. net/), and PinDel (http://trac.nbic.nl/pindel/). Routine genotyping of mice after identification of the integration site was conducted by multiplex PCR using a primer set (*Chr_BP1-F*: 5′-GATGGGCCA CACTCTTAGAGCCTCTG-3′, *Insert_BP1-R*: 5′-AGCGCAGAAGTGGTC CTGCAACTT-3′ and Del_BP1-R: 5′-GGAGTTTGTTTGGGTCTTTTGT TGCTTG-3′) that produced 503- and 411-bp fragments from the Tg-integrated and normal alleles, respectively.

### Behavioral analysis

PPI scores and ASR against acoustic stimuli in mice were measured according to previously published methods (Shimamoto *et al*, 2014). Sodium hydrosulfide (NaHS) was purchased from Nacalai Tesque (Kyoto, Japan). This compound was dissolved in saline and administered to a mouse intraperitoneally at a dose of 1 mg/kg or 10 mg/kg body weight once a day. Control mice were administered saline.

### Real-time quantitative RT–PCR and digital RT–PCR

Single-stranded cDNA for each total RNA sample was synthesized using SuperScript III RT (Invitrogen), oligo(dT), and random hexamers (Maekawa *et al*, 2015). Real-time quantitative PCR analysis was conducted using an ABI7900HT Fast real-time PCR system (Applied Biosystems). The TaqMan probes used were TaqMan™ Gene Expression Assay products (Applied Biosystems). All real-time quantitative PCR data were captured using SDS v2.4 (Applied Biosystems). The ratio of the relative concentration of the target molecule to that of the *GAPDH* gene (target molecule/*GAPDH* gene) was calculated. All reactions were performed in triplicate based on a standard curve method. Outliers (more or less than mean ± 2SD) were excluded.

Absolute quantitative analysis of mRNAs was conducted using the QuantStudio™ 3D digital PCR system (ThermoFisher, Waltham, MA, USA). The TaqMan probes used were TaqMan™ Gene

Expression Assay products (Applied Biosystems). All absolute quantitative data were captured using QuantStudio™ 3D AnalysisSuite™ software (ThermoFisher).

### Human samples

A set of RNA samples derived from BA8 was obtained from the Victorian Brain Bank Network (http://www.mhri.edu.au/brain-bank) (1st set). Demographic data of the BA8 samples are described in Appendix Table S7. Another set of postmortem brain tissues (BA17) from schizophrenia and age-matched controls were obtained from the Postmortem Brain Bank of Fukushima for Psychiatric Research and Brain Research Institute, Niigata University, Japan (in total *n* = 22 for schizophrenia and *n* = 14 for control) (2nd set) (Ohnishi *et al*, 2019) (Appendix Table S8).

Characteristics of the human iPS cells and neurosphere samples analyzed are described elsewhere (Bundo *et al*, 2014; Maekawa *et al*, 2015; Toyoshima *et al*, 2016). Peripheral whole-blood samples were collected, and total RNA was extracted using the RiboPure™-Blood Kit (Ambion, Grand Island, NY, USA). MPST levels in plasma were measured using a human ELISA kit for 3-mercaptopyryvate sulfurtransferase (MST) (Wuhan USCN Business, Hubei, China). Subjects consisted of 44 subjects with schizophrenia and 56 control subjects, all from the Tokyo area of Japan. Diagnoses were made in accordance with the DSM-IV criteria. Demographic data are shown in Appendix Table S8.

Preparation of scalp hair follicle samples and the method used for RNA extraction from those samples are described elsewhere (Maekawa *et al*, 2015). We collected 10–12 hair follicles from an individual, with RNA yield per one hair follicle being 58.09 ± 27.61 ng (mean ± SD). We did not amplify RNA but pre-amplified cDNA of all the target genes before quantification, using TaqMan PreAmp Master Mix (ThermoFisher). Demographic data are shown in Appendix Table S9.

The human studies, including the use of iPS cells, conformed to the principles set out in the WMA Declaration of Helsinki and the NIH Belmont Report and were approved by the Ethics Committees of all relevant institutes, and all the participants provided written informed consent to participate in the study.

### Immunohistochemistry and *in situ* hybridization of MPST/*MPST* in hair follicles

Analysis of MPST protein expression in scalp hair follicle samples was performed as previously reported (Maekawa *et al*, 2015, 2019). Primary polyclonal rabbit antibody was used for MPST detection (Santa Cruz Biotechnology (Dallas, TX, USA); dilution 1:100). The secondary antibodies used were Alexa Fluor 488- or 594-labeled goat anti-rabbit IgG or goat anti-guinea pig IgG (Life Technologies; dilution, 1:400).

Sequences for *in situ* probes were amplified by PCR using the appropriate primers and cloned into pGEM-T (Promega, Fitchburg, WI, USA). Digoxigenin (DIG)-labeled RNA probes were prepared with a template plasmid and the DIG RNA Labeling Kit (Roche, Basel, Switzerland). After drying, the sections were fixed for 10 min in 4% paraformaldehyde in PBS at room temperature. The sections were rinsed with PBS and incubated with 0.5 μg/ml proteinase K (in 10 mM Tris–HCl, pH 7.4, 1 mM EDTA) for 5 min at 37°C. Fixing was

repeated with 4% paraformaldehyde in PBS for 10 min, followed by rinsing with PBS. Probes were diluted (1:200) with hybridization buffer [5× SSC, 50% formamide, 1% (v/v) SDS, 50 μg/ml heparin, 50 μg/ml tRNA], and 300 μl of each sample was applied to a slide. After 16 h of incubation at 65°C, the sections were washed, first with MABT buffer [100 mM maleic acid, pH 7.5, 150 mM NaCl and 0.3% (v/v) Tween 20] twice for 5 min, then with washing buffer [1× SSC, 50% formamide, 0.1% (v/v) Tween 20] twice for 30 min at 65°C, and finally with MABT buffer twice for 5 min. Then, the slides were incubated in PBS containing 3% $H_2O_2$ for 30 min and washed and blocked with blocking reagent (Roche) for 1 h. After blocking, the slides were incubated with HRP-conjugated anti-DIG antibody (1:500; Roche) in blocking reagent for 16 h at 4°C. The samples were washed three times (5 min each) with MABT buffer. Signals were visualized by incubating the sample with Tyramide-Alexa Fluor 488 (1:200) for 10 min using a TSA kit (Life Technologies). After three rinses with PBS, coverslips were placed on the slides for examination. Fluorescence signals were detected using an FV1000 confocal laser-scanning microscope (Olympus, Tokyo, Japan).

**Western blot analysis of MPST in postmortem brain samples**

Western blot analysis was performed as per our previous report (Ohnishi et al, 2019). The protein (40 μg/lane) was analyzed in Western blot with anti-MPST (Nagahara et al, 2013; dilution 1:3,000) and anti-GAPDH (Santa Cruz Biotechnology, sc-20357; dilution 1:5,000) antibodies. The chemiluminescent intensities for the signal of MPST were normalized to those of GAPDH.

**Receiver operating characteristic (ROC) curve analysis**

The equations for calculation of sensitivity and specificity were "sensitivity = number of true positives/(number of true positives + number of false negatives)" and "specificity = number of true negatives/(number of false positives + number of true negatives)" with the criterion that relative expression levels of *MPST* mRNA above a cut-off value were deemed as positive for schizophrenia.

We used the Mann–Whitney *U* test (two-tailed) to detect significant changes in the expression levels of each gene. Correlations of confounding factors for *MPST* gene expression were calculated by Spearman's rank correlation test.

**RNA-seq analysis**

Transcriptome analysis in the frontal cortical brain region was performed by RNA-seq in (a) *Mpst*-KO mice (n = 6) versus WT controls (n = 6) and (b) *Mpst*-Tg (n = 6) versus littermate (non-Tg) controls (n = 6). Total RNA from the frontal cortical brain region was extracted (miRNeasy Mini Kit, Qiagen, Hilden, Germany), and the quantity and quality of RNA were estimated (2200 TapeStation system, Agilent, Santa Clara, CA, USA). Samples with RNA integrity number (RIN) ≥ 8.7 were further used for library preparation with 200 ng of total RNA according to the manufacturer's instructions for the TruSeq Stranded mRNA Sample Prep Kit (Illumina, San Diego, CA, USA). Briefly, poly-A-containing mRNA was purified using poly-T oligo-attached magnetic beads and then heat fragmented and reverse transcribed into first-strand cDNA using reverse transcriptase and random primers. Second-strand cDNA synthesis was performed

by incorporating dUTP followed by addition of a single "A" nucleotide at the 3′ ends of the blunt fragments to prevent ligation of double-stranded cDNA. After adapter ligation (including multiplexing barcodes), the cDNA fragments were enriched by PCR (15 cycles) to create the final cDNA library. The quality, size distribution, and quantity of the cDNA libraries were assessed (2100 Bioanalyzer, Agilent), and the libraries were further sequenced in 100-bp paired-end read format on the HiSeq 2500 platform (Illumina).

The RNA-seq data for the individual samples were demultiplexed using the unique index adapters. The quality of the sequence reads was evaluated by FastQC (http://wwwbioinformaticsbabrahamacuk/projects/fastqc), and the reads were trimmed for adapter sequences and low-quality bases using the FASTX tool kit (http://hannonlabcshledu/fastx_toolkit/indexhtml). The reads were further mapped to the mouse reference genome (GRCm38/mm10, http://hgdownload.soe.ucsc.edu/goldenpath/mm10/chromosomes/) using TopHat with default parameters (v.2.0.14) (Trapnell et al, 2012), utilizing the aligner Bowtie2 (v.2.2.5) (Langmead et al, 2009). The expression levels were quantified using Cufflinks (v.2.2.1) (Trapnell et al, 2012) based on the read mapping and calculated as fragments per kilobase of transcript per million mapped reads (FPKM), corresponding to the UCSC gene annotations for mm10 (http://hgdownload.soe.ucsc.edu/goldenpath/mm10/database/refFlat.txt.gz). To test the statistical significance for differential expression among the comparison groups, Student's *t*-test on log-transformed FPKM values ($\log_2$ FPKM) was applied. *P* values < 0.05 were considered statistically significant, and these reads were further analyzed for gene ontology enrichment and pathway analysis. Visualization of differentially expressed genes using volcano plots was performed in R (https://www.r-project.org). Differentially expressed genes were tested for gene ontology enrichment and pathway analysis. Gene ontology enrichment analysis was performed in the PANTHER Overrepresentation Test (http://pantherdb.org/webservices/go/overrep.jsp, annotation version and release date: GO Ontology database, released on 2018-06-01). Reported *P* values were corrected for multiple testing using the FDR method. Pathway enrichment was performed using Ingenuity Pathway Analysis (IPA) (Qiagen, content version: 36601845, release date: 2017-06-22). The statistical significance of the enriched canonical signaling pathways was calculated using Fischer's exact test. *P* values < 0.05 were considered statistically significant.

**Analysis of metabolites**

To minimize postmortem degradation, we subjected mice to high-energy focused beam microwave irradiation (5 kW, 0.94 s; Muromachi Kikai, Tokyo, Japan) without anesthesia and then analyzed the extracted brains (Sugiura et al, 2014). A report provided by the American Veterinary Medical Association Recommendations described that microwave irradiation is a humane method for euthanizing small laboratory rodents, with the advantages that unconsciousness is achieved in < 100 ms and a complete loss of brain function in < 1 s.

ATP and ADP concentrations were measured using MRM (Multiple Reaction Monitoring) method by the LC-triple quadrupole mass spectrometer (UHPLC: ACQUITY UPLC, Waters, Milford, Massachusetts, USA; mass spectrometer: TSQ Vantage EMR, ThermoFisher), and using d5-AMP as an internal standard, at the Support Unit for Bio-Material Analysis in RIKEN Center for Brain

Science, Research Resource Division (RRD). Mouse samples (more or less than mean ± 1.5SD in terms of ATP-to-ADP ratio) were excluded ($n = 1$ for non-Tg group and $n = 1$ for *Mpst*-Tg group) for analysis.

### Analysis of mitochondrial cytochrome c oxidase activity

Mitochondrial cytochrome c oxidase activity was measured in non-Tg ($n = 6$) and *Mpst*-Tg mice ($n = 6$) (both age of 6 weeks). Fresh mitochondria were isolated from cerebral hemisphere by using Mitochondria Isolation Kit for Tissue and Cultured Cells (Biochain, Newark, CA, USA) and collected pellets were stored at −80°C until use. At the timing of measurement, pellets were re-suspended in the storage buffer [10 mM HEPES (pH 7.1–7.5), 250 mM sucrose, 1 mM ATP, 80 µM ADP, 5 mM sodium succinate, and 2 mM $K_2HPO_4$]. The activity was determined by Cytochrome c Oxidase Assay Kit (Sigma). The assay was performed using Ultrospec 3300 pro (GE Healthcare). Results were represented in units per mg of protein defined by Micro BCA Protein Assay Kit (ThermoFisher).

### Analysis of spines

Primary culture of E16.5 hippocampal neurons from non-Tg and *Mpst*-Tg mice were performed as previously described (Morikawa *et al*, 2018). Briefly, dissociated hippocampal neurons were plated on chambered coverslips coated with polyethyleneimine at $1 \times 10^5$ cells per well and cultured in a 5% $CO_2$ atmosphere at 37°C. Cultured hippocampal neurons at DIV 14–21 were transfected with an EGFP expression vector using the calcium phosphate method for 2 days before the observation. The dendritic spines were observed using a confocal laser-scanning microscope (LSM780) equipped with an Airyscan module (ZEISS). Data analyses were performed using IMARIS (BITPLANE) software (Oxford Instruments, Abingdon, UK).

### Protein S-palmitoylation assay

Protein *S*-palmitoylation in the brain was examined by the modified ABE method (Forrester *et al*, 2011) with minor modifications. Brain tissues (~20 mg wet weight) were lysed in 250 µl of lysis buffer (50 mM Tris–HCl (pH 7.5), 150 mM NaCl, 5 mM EDTA, 4% SDS, 2 mM PMSF) by sonication. The lysates were diluted with 250 µl of Triton buffer (50 mM NaCl (pH 7.5), 150 mM NaCl, 5 mM EDTA, 0.2% Triton X-100, 2 mM PMSF) and then centrifuged for 10 min at 18,000 *g* and 15°C. The supernatants (~1.1 mg of protein) were treated with 20 mM TCEP [Tris(2-carboxyethyl)phosphine, FUJI-FILM Wako Pure Chemical Co., Osaka, Japan] for 30 min at room temperature in a reaction volume of 1 ml to cleave disulfide bonds. Next, the mixtures were treated with 40 mM NEM (N-ethyl maleimide) for 2.5 h to block free thiol groups. For detection of "global" *S*-palmitoylation, one-tenth of the mixture was stored as "input" for Western blot analysis with HRP-labeled streptavidin (GE Healthcare) under non-reducing conditions followed by non-reducing SDS–PAGE. For detection of palmitoylation of specific proteins, the remaining mixture after the ABE reaction was extracted with 4 ml of methanol, 1 ml of chloroform, and 3 ml of water, and the upper phase obtained after centrifugation at 5,800 *g* for 10 min was removed. The pellet was rinsed twice with 10 ml of methanol. The pellets were completely dissolved in 550 µl of

4% SDS buffer (4% SDS, 50 mM Tris–HCl (pH 7.5), 5 mM EDTA) by sonication. To a 250-µl aliquot, 750 µl of HA(+) buffer [1 M hydroxyamine (pH 7.5), 150 mM NaCl, 0.2% Triton X-100, 1 mM biotin-HPDP (*N*-[6-(biotinamide)hexyl]-3′-(2′-pyridyldithio)propionamide; ThermoFisher), and 1 mM PMSF] or HA(−) buffer (1 M Tris–HCl (pH 7.5), 150 mM NaCl, 0.2% Triton X-100, 1 mM biotin-HPDP, and 1 mM PMSF). After incubation for an hour, the mixture was extracted with 4 ml of methanol, 1 ml of chloroform, and 3 ml of water, and the upper phase obtained after centrifugation at 5,800 *g* for 10 min was removed. The pellet was rinsed twice with 10 ml of methanol. The pellet was dried and then dissolved in 170 µl of 2% SDS buffer [2% SDS, 50 mM Tris–HCl (pH 7.5), 150 mM NaCl, 5 mM EDTA, and 1 mM PMSF] by sonication. After diluting with 10 volumes of 0.2% Triton buffer (0.2% Triton X-100, 50 mM Tris–HCl (pH 7.5), 5 mM EDTA), the biotinylated proteins were captured with NeurAvidin agarose beads (ThermoFisher). The complex was rinsed with wash buffer (0.2% Triton buffer supplemented with 0.1% SDS) three times. Biotinylated proteins were released with 1.5× SDS sample buffer by boiling for 10 min and subjected to Western blot analysis with an anti-PSD95 (Cell Signaling, Danvers, MA, USA; cat. no. 3450), anti-Ras (Cell Signaling, cat. no. 3965), anti-Cdc42 (Cell Signaling, cat. no. 2466), or anti-MBP (Cell signaling, cat. no. 78896) antibody.

### Genetic association study

The genetic association of *MPST* and *CBS* with schizophrenia was evaluated by analyzing SNPs in 2,011 cases and 2,170 controls (Bangel *et al*, 2015; Balan *et al*, 2017). *MPST* was examined using 4 tag SNPs, and *CBS* was assessed with 7 tag SNPs and two promoter SNPs (rs1788484 and rs2850144) (Appendix Table S15). The promoter SNPs are described in the Tohoku Medical Megabank Organization of Tohoku University (ToMMo) database of the Japanese population (https://ijgvd.megabank.tohoku.ac.jp) (Nagasaki *et al*, 2015). The SNPs were genotyped using TaqMan SNP genotyping assays (Applied Biosystems) following the manufacturer's instructions.

### DNA methylation analysis

DNA methylation was performed using EpiTYPER on the MassARRAY System platform (Agena Bioscience, San Diego, CA, USA) following the manufacturer's instructions. The sequence information of the probes used for DNA methylation analysis and information on the lengths of PCR products is shown in Appendix Table S17.

### Poly-I:C model

Pregnant B6 mice received five consecutive intraperitoneal injections of poly-I:C (2 mg/ml, Sigma) dissolved in PBS (20 mg/kg) or an equivalent volume of PBS at embryonic days 12, 13, 14, 15, and 16. At adulthood (13–14 weeks old), brains were dissected from pups for analysis.

## Data availability

RNA-seq data of *Mpst*-KO and Tg mice are available on NCBI BioProject (https://www.ncbi.nlm.nih.gov/bioproject) accession

## The paper explained

### Problem

Schizophrenia is a severe mental illness, and susceptibility to schizophrenia is given by multiple genetic variants and environmental factors, the latter particularly being subtle insults including oxidative/inflammatory stress during the early brain development (termed the "neurodevelopmental hypothesis"). However, a precise mechanism for non-genetic effects remains largely elusive.

### Results

We focused on the differential integrity of prepulse inhibition (PPI), a representative endophenotype of schizophrenia, between different inbred mouse strains. Our proteomics analysis and examination of gene-manipulated mice revealed that elevated levels of Mpst, a hydrogen sulfide ($H_2S$)/polysulfides-producing enzyme, led to impaired PPI with increased sulfide deposition. Analysis of human samples demonstrated that the $H_2S$/polysulfides production system is indeed upregulated in schizophrenia ("sulfide stress"). Mechanistically, *Mpst* overexpression dampened energy metabolism, while maternal immune activation model mice showed upregulation of genes for $H_2S$/polysulfides production, partly via epigenetic modifications. These results suggest that inflammatory/oxidative insults in early brain development result in upregulated $H_2S$/polysulfides production as an antioxidative response, which in turn cause deficits in bioenergetic processes.

### Impact

This study revealed that an elevated $H_2S$/polysulfides-producing system ("sulfide stress") underlies the pathophysiology of a subset of schizophrenia, explaining a novel aspect of the "neurodevelopmental hypothesis", and could provide a novel paradigm for drug development.

numbers PRJDB8817 and PRJDB8818, respectively. Proteomics data of mouse brain and lymphocytes are available on PeptideAtlas (http://www.peptideatlas.org) Dataset Identifier PASS01452.

Expanded View for this article is available online.

## Acknowledgements

We deeply thank Yuko Fukata (National Institute for Physiological Sciences) for valuable suggestions regarding the palmitoylation assay. We are grateful to Noriyuki Nagahara (Nippon Medical School) for kindly providing the anti-Mpst polyclonal antibody (Nagahara *et al*, 1998). pCAGGS (Niwa *et al*, 1991) was provided by the RIKEN BRC through the National BioResource Project of the MEXT/AMED, Japan. We also thank members of Research Resources Division, RIKEN CBS, for animal maintenance, embryo manipulation, metabolite analysis and DNA sequencing service, Takashi Asada for his help for the establishment of collaborative team, Miyuki Kato and Santha Kumara Dissanayaka for technical assistance, Atsuko Nagata, Junya Matsumoto and Mizuki Hino for the preparation of postmortem brain samples, Tomoe Ichikawa, Kazuya Toriumi and Akiko Kobori for their help for collecting scalp hair follicle samples, Masaomi Iyo and Toshihisa Niitsu for collecting plasma samples, Tadayuki Ogawa for his help for spine analysis, and Makoto Asashima and Renpei Nagashima for their helpful comments and discussions. This study was supported by the Strategic Research Program for Brain Sciences from Japan Agency for Medical Research and Development (AMED) under Grant Numbers JP18dm0107083 (TY), JP19dm0107083 (TY), JP18dm0107085 (HK), JP19dm0107119 (KH), JP19dm0908001 (NH), 19dm0107086 (YKu), and 19dm0107107 (HY), and by the Grant-in-Aid for Scientific Research on Innovative Areas from the MEXT under Grant Numbers JP18H05435 (TY), JP18H05428 (TY), and JP16H06277 (HY), and by JSPS KAKENHI under Grant Number JP17H01574 (TY). This study was also supported in part by Grants-in-Aid for NEDO (New Energy and Industrial Technology Development Organization) (KU). This work was performed in part as a collaborative research effort of Clinical Bioinformatics Research Initiative (CBIRI) at the National Institute of Advanced Industrial Science and Technology (AIST), Japan.

## Author contributions

Conception and design: MId, KU, TY; mass spectrometry analysis: TI, KM, TKata, KU; biochemical analysis of sulfides: NS, YKi, HK; experiments using animals and human samples (generation of model animals, behavioral analysis, genetic analysis, biochemical analysis, metabolic analysis, spine analysis, etc.): TO, MT, MMa, C-SM, YI, HOh, AW, YH, YM, TH, MMo, KH, YN, YW, YT, TKato, AN, SF, NH, KI; iPS cell study: MT, YH, HOk; collection and management of postmortem brain samples: YKu, AK, HY, BD; collection and evaluation of other clinical samples: KH, TT, MIt; analysis and interpretation of data (e.g., statistical analysis, biostatistics, computational analysis): MId, TO, MT, SB, YI, AW, TH, HK, TY; writing, review, and/or revision of the manuscript: MId, TO, SB, BD, HK, TY; study supervision: TY.

## Conflict of interest

The authors declare that they have no conflict of interest.

## For more information

(i) The novel insertion/deletion polymorphism in the *Npm1* gene and a novel missense polymorphism in the *Nme2* gene in mouse genome, which we have identified in this study, have been deposited in the NCBI database (http://www.ncbi.nlm.nih.gov) and have been assigned the tentative IDs ss410758760 and ss410758759, respectively.

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
