## [Review Process File · EMBO Molecular Medicine]

Excess hydrogen sulfide and polysulfides production underlies a schizophrenia pathophysiology

Masayuki Ide, Tetsuo Ohnishi, Manabu Toyoshima, Shabeesh Balan, Motoko Maekawa, Chie Shimamoto-Mitsuyama, Yoshimi Iwayama, Hisako Ohba, Akiko Watanabe, Takashi Ishii, Norihiro Shibuya, Yuka Kimura, Yasuko Hisano, Yui Murata, Tomonori Hara, Momo Morikawa, Kenji Hashimoto, Yayoi Nozaki, Tomoko Toyota, Yuina Wada, Yosuke Tanaka, Tadafumi Kato, Akinori Nishi, Shigeyoshi Fujisawa, Hideyuki Okano, Masanari Itokawa, Nobutaka Hirokawa, Yasuto Kunii, Akiyoshi Kakita, Hirooki Yabe, Kazuya Iwamoto, Kohji Meno, Takuya Katagiri, Brian Dean, Kazuhiko Uchida, Hideo Kimura and Takeo Yoshikawa.

Review timeline:

Submission date:	2 nd May 2019
Editorial Decision:	4 th June 2019
Revision received:	30 th August 2019
Editorial Decision:	17 th September 2019
Revision received:	25 th September 2019
Accept:	1 st October 2019

Editor: Celine Carret

Transaction Report:

1st Editorial Decision

4th June 2019

Thank you for the submission of your manuscript to EMBO Molecular Medicine. We have now heard back from the three referees whom we asked to evaluate your manuscript.

Overall, you will see that the referees find the paper to be interesting and clinically relevant. While ref. #1 and #3 are supportive of publication with only minor (mostly text changes) revision, ref. 2 is a little bit more critical and would like to see some mechanism, along with better discussions and clarification of the main message. At this point, and given that the study already offers a great amount of work, I would strongly recommend you to add mechanistic data if you have some at hand. In any case, please focus on the main message of the article and answer all other comments adequately.

We would therefore welcome the submission of a revised version within three months for further consideration and would like to encourage you to address all the criticisms raised as suggested to improve conclusiveness and clarity.

***** Reviewer's comments *****

Referee #1 (Comments on Novelty/Model System for Author):

The authors propose a novel idea for the pathogenic role of H₂S/polysulfides in schizophrenia (SCZ). The idea evolved from investigations into the underlying basis for contrasting prepulse inhibition (PPI) levels in 2 inbred mouse strains. PPI is a proposed endophenotype in SCZ. Analysis

was thorough and dissemination of these findings in SCZ would stimulate replication in orthogonal samples.

Referee #1 (Remarks for Author):

Ide and colleagues report on a not-very-commonly-thought-of-idea that involves the potential role of sulfur-containing compounds in prepulse inhibition (PPI), a proposed endophenotype of schizophrenia (SCZ), and in SCZ, itself. The authors report that two inbred mouse strains displayed contrasting PPIs, B6 showed the highest PPI while C3H had the lowest PPI, analogous to the PPI deficiency associated with SCZ patients. Proteomic analysis in brain implied that the divergent PPIs correlated with the expression levels of Mpsst, H₂S/polysulfide producing enzyme; C3H and B6 carry elevated and low levels of the enzyme, respectively. Mpsst knockout in C3H reduced sulfur storage and improved PPI whereas B6 Mpsst transgene showed impaired PPI. SCZ postmortem brains had increased expression of MPST and CBS, a gene that codes for another enzyme involved in H₂S formation. Similarly, hair follicles in SCZ patients showed greater expression of MPST than in controls and iPSC-derived neurospheres from 22q deletion SCZ patients showed higher CBS mRNA than in controls. The authors propose a pathogenic role of "sulfide stress" in a subset of SCZ.

Comments

- The proposed role of H₂S/polysulfides in SCZ is an interesting novel idea that needs to be verified in orthogonal samples, preferably in SCZ patients that show divergent levels of PPI.
- This is a very well-designed study that examined an endophenotype of SCZ in inbred mice with contrasting PPI levels, evidence for its underlying cause is presented, i.e., increased H₂S/polysulfide formation due to high levels of Mpsst.
- Extending the investigation to SCZ-derived tissue samples, the authors found support for the potential pathogenic role of H₂S/polysulfides.
- The authors performed extensive studies in mice, creating knockout and transgenic models in the appropriate mouse strain.
- Postmortem brain analysis and SCZ iPSC-derived neurospheres reinforced findings generated from mouse studies.

Specific comments

- In the text describe briefly the hair follicle method, indicating the number of patients and controls that donated hair follicles for the study, the number of hair follicles taken from each individual and the total RNA yield per number of hair follicles and whether amplification of RNA was performed prior to analysis.
- Do any of the genes that encode the sulfide/sulfane forming enzymes map to the region of the top association hits for schizophrenia?
- Any possible alternative reasons for the PPI difference between B6 and C3H other than that proposed in this report?
- Describe the methylation experiment more clearly, e.g., explain the source for the "CpG227" probe.
- "Sulfide content in the brain was higher in B6 mice than in C3H mice". This is the heading of the section on page 8. Is this correct? Rather, should it read "Sulfide content in the brain was higher in C3H mice than in B6 mice"?

Referee #2 (Comments on Novelty/Model System for Author):

The topic of the submission is clearly interesting and it is obvious that a lot of work and state-of-the-art techniques have gone into this project. However, there are several potential issues that need to be addressed:

MAJOR

- The overall 'message' of the paper is unclear as far as the role of H₂S and polysulfides is concerned. In some parts, it seems that the authors regard the upregulated H₂S pathways as responses to an earlier (perhaps perinatal) oxidative stress; in other parts it seems that the authors

regard these pathways as causative for schizophrenia (e.g. the fact that NaHS donation on its own can induce some features of the disease). These concepts should be better integrated into a holistic concept.

- It is unclear what enzyme (3-MST or CBS or both?) seems to play a role. In different assays (brain tissues, in vitro engineered neurospheres, hair follicle-derived materials), sometimes it is CBS, other times it is 3-MST. This appears to be confusing. Also, when it comes to human tissues, it is unclear why authors did not study brain parts that are traditionally considered more relevant for schizophrenia (temporal lobe, frontal cortex? In fact in the supplemental data it is shown that the frontal cortex shows no differences in 3MST and CBS expression?)

- The extent of the changes in 3-MST or CBS expression, in some of the experiments, appears to be minimal (perhaps a 5% change from control; Fig 6A, 6B, 6I; Fig S10H-I). First of all, one should subject the data to an independent statistical review to ascertain that the analysis is correct. Second, even if the difference is statistically detectable, it is questionable whether such small changes are biologically relevant in human disease.

- Some of the animal data do not fully support the authors' conclusions. For example, the difference between the two strains in ASR responses (Fig 4B) is large but the effect of 3-MST deficiency or 3-MST overexpression on the same response (Fig 4C, 4E) is minimal. These data would indicate that the majority of the factors that explain the strain difference are actually not related to the 3-MST pathway but to other factors.

- No mechanism is studied or proposed (at least directly; the ingenuity pathway analysis is only correlative) as far as the actual mechanism of the excess H₂S is concerned. How does the extra H₂S promote the pathogenesis of schizophrenia? Much more direct experimental work would be needed to define this point.

- While the experiments with PolyIC are interesting (on their own), and they do demonstrate that the polyIC exposure CAN affect H₂S-pathway-related pathways, this experiment does not actually prove or demonstrate that this is, indeed the case for schizophrenia patients or for the strain difference that the entire project started out with. Is it known that one strain of mice has perinatal immune stimulation or oxidative stress while the other does not?

MINOR

The CPG methylation data are correlative but do not prove a mechanistic relation; in order to prove this, additional data would be needed.

If 3-MST is a H₂S producing enzyme, how is it possible that plasma H₂S levels are not different between WT and 3-MST KO mice or WT and 3-MST overexpressors?

Referee #2 (Remarks for Author):

The topic of the submission is clearly interesting and it is obvious that a lot of work and state-of-the-art techniques have gone into this project. However, there are several potential issues that need to be addressed:

MAJOR

- The overall 'message' of the paper is unclear as far as the role of H₂S and polysulfides is concerned. In some parts, it seems that the authors regard the upregulated H₂S pathways as responses to an earlier (perhaps perinatal) oxidative stress; in other parts it seems that the authors regard these pathways as causative for schizophrenia (e.g. the fact that NaHS donation on its own can induce some features of the disease). These concepts should be better integrated into a holistic concept.

- It is unclear what enzyme (3-MST or CBS or both?) seems to play a role. In different assays (brain tissues, in vitro engineered neurospheres, hair follicle-derived materials), sometimes it is CBS, other

times it is 3-MST. This appears to be confusing. Also, when it comes to human tissues, it is unclear why authors did not study brain parts that are traditionally considered more relevant for schizophrenia (temporal lobe, frontal cortex? In fact in the supplemental data it is shown that the frontal cortex shows no differences in 3MST and CBS expression?)

- The extent of the changes in 3-MST or CBS expression, in some of the experiments, appears to be minimal (perhaps a 5% change from control; Fig 6A, 6B, 6I; Fig S10H-I). First of all, one should subject the data to an independent statistical review to ascertain that the analysis is correct. Second, even if the difference is statistically detectable, it is questionable whether such small changes are biologically relevant in human disease.

- Some of the animal data do not fully support the authors' conclusions. For example, the difference between the two strains in ASR responses (Fig 4B) is large but the effect of 3-MST deficiency or 3-MST overexpression on the same response (Fig 4C, 4E) is minimal. These data would indicate that the majority of the factors that explain the strain difference are actually not related to the 3-MST pathway but to other factors.

- No mechanism is studied or proposed (at least directly; the ingenuity pathway analysis is only correlative) as far as the actual mechanism of the excess H₂S is concerned. How does the extra H₂S promote the pathogenesis of schizophrenia? Much more direct experimental work would be needed to define this point.

- While the experiments with PolyIC are interesting (on their own), and they do demonstrate that the polyIC exposure CAN affect H₂S-pathway-related pathways, this experiment does not actually prove or demonstrate that this is, indeed the case for schizophrenia patients or for the strain difference that the entire project started out with. Is it known that one strain of mice has perinatal immune stimulation or oxidative stress while the other does not?

MINOR

The CPG methylation data are correlative but do not prove a mechanistic relation; in order to prove this, additional data would be needed.

If 3-MST is a H₂S producing enzyme, how is it possible that plasma H₂S levels are not different between WT and 3-MST KO mice or WT and 3-MST overexpressors?

Referee #3 (Comments on Novelty/Model System for Author):

This manuscript used state-of-the art methodologies.

A novel pathophysiology that is certainly important for a subset of patients with schizophrenia is evidenced.

This manuscript has a high medical impact potential.

Mouse models (KO and overexpression model) are adequate.

Referee #3 (Remarks for Author):

Manuscript 'Excess hydrogen sulfide and polysulfides production underlies a schizophrenia pathophysiology'

The authors have already reported that, across 4 strains of mice, C57BL/6N (B6) mice exhibited the highest prepulse inhibition (PPI) scores while C3H/HeN (C3H) the lowest (Watanabe et al, 2007). Prepulse inhibition (PPI) of the startle reflex has been suggested as a candidate endophenotype for schizophrenia research, as it shows high heritability and has been found deficient in schizophrenia spectrum disorders (Roussos et al, 2016).

Furthermore, Watanabe et al., 2007 were able to identify Fabp7 as Quantitative Trait Locus for a

Schizophrenia Endophenotype . Fabp7 (fatty acid binding protein 7, brain) is a gene with functional links to the N-methyl-D-aspartic acid (NMDA) receptor.

The manuscript of Ide et al. presents a convincing set of results, based on state-of-the-art methodologies to demonstrate that prepulse inhibition (PPI) endophenotypes levels depends on hydrogen sulfide and polysulfides production.

The manuscript is articulated on (i) proteomics studies of prefrontal cortex between the two inbred strains, (ii) the analysis of Mpst and genes encoding the other H₂S-producing enzymes and profile of H₂S metabolic states in mice, (iii) the study of mouse behaviors and sulfide deposition in Mpst KO and Mpst-Tg mice, (iv) the expression levels of genes encoding H₂S-producing enzymes and between DNA methylation levels and gene expression levels, (v) H₂S-producing genes/proteins in human brain-related samples, (vi) the RNA-seq analysis of frontal cortex/ Mpst KO and Tg mice and (vii) Gene expression in the maternal immune activation mouse model.

Altogether, these results are robust and underscore the role of an excess hydrogen sulfide and polysulfides production as able to generate a schizophrenia endophenotype.

This manuscript deserves to be published in EMBO Molecular Medicine with minor revision.

Minor comments:

Abstract: abstract needs to be modified, tacking in account "The paper explained" with emphasis on "neurodevelopmental hypothesis" and analysis of a maternal immune activation mouse model for schizophrenia, that are not mentioned in the abstract.

Introduction:

As EMBO Molecular Medicine is not a psychiatry specialized journal, "maternal immune activation mouse model for schizophrenia" and "neurodevelopmental hypothesis of schizophrenia" need to be introduced here.

Some general references (i.e. Knuesel I, Chicha L, Britschgi M, Schobel SA, Bodmer M, Hellings JA, Toovey S, Prinssen EP. Maternal immune activation and abnormal brain development across CNS disorders. *Nat Rev Neurol*. 2014 Nov;10(11):643-60; Myka L. Estes and A. Kimberley McAllister. Maternal immune activation: Implications fo rneuropsychiatric disorders, *Science* 2016) need to be included here.

Results:

Excess H₂S/polysulfides production elicits dampened expression of genes for energy metabolism (p12)

The authors indicate: "Interestingly, the dysregulated genes in the Mpst Tg mice were significantly enriched in the glutamatergic synaptic transmission and synaptic signaling-related ontology terms, along with the cellular metabolic processes (Fig 8C and Supplementary Table S11)". I suggest to present these results in more details as they are related to "neurodevelopmental hypothesis" and previous results on PPI, identify Fabp7 and NMDA receptors. Similarly, "axonal guidance" pathway needs to be discussed in regard of cortical long-range projection pathways that are involved in schizophrenia.

Inflammatory/oxidative stress in the early brain developmental stage results in upregulated H₂S/polysulfides production (p15-16)

Fold changes for Sod1 and Sod2 (encoding superoxide dismutase), Cat (encoding catalase) and Gpx1 and Gpx4 (encoding glutathione peroxidase need to be indicated and discussed (Fig. 9E-I). In spite of the statistically significant changes evidences, fold-changes are quite small and this point needs to be emphasized.

Discussion (p16-19)

"maternal immune activation mouse model for schizophrenia" and "neurodevelopmental hypothesis od schizophrenia " need to be discussed here.

The phenotype of Cbs overexpression mouse line that displays Novel Object recognition (NOR) defects more in relation with Intellectual Disability (ID) than schizophrenia (Marechal D, Brault V, Leon A, Martin D, Lopes Pereira P, Loaëc N, Birling MC, Friocourt G, Blondel M, Herault Y. Cbs overdosage is necessary and sufficient to induce cognitive phenotypes in mouse models of Down syndrome and interacts genetically with Dyrk1a. *Hum Mol Genet*. 2019 May 1;28(9):1561-1577). These novel results need to be discussed.

Legends of Figures

The legends need to be improved (i.e. p. 56, Fig 9 E-I: no information is given).

Response to the comments of reviewer #1:

The authors propose a novel idea for the pathogenic role of H₂S/polysulfides in schizophrenia (SCZ). The idea evolved from investigations into the underlying basis for contrasting prepulse inhibition (PPI) levels in 2 inbred mouse strains. PPI is a proposed endophenotype in SCZ. Analysis was thorough and dissemination of these findings in SCZ would stimulate replication in orthogonal samples.

Ide and colleagues report on a not-very-commonly-thought-of-idea that involves the potential role of sulfur-containing compounds in prepulse inhibition (PPI), a proposed endophenotype of schizophrenia (SCZ), and in SCZ, itself. The authors report that two inbred mouse strains displayed contrasting PPIs, B6 showed the highest PPI while C3H had the lowest PPI, analogous to the PPI deficiency associated with SCZ patients. Proteomic analysis in brain implied that the divergent PPIs correlated with the expression levels of Mpst, H₂S/polysulfide producing enzyme; C3H and B6 carry elevated and low levels of the enzyme, respectively. Mpst knockout in C3H reduced sulfur storage and improved PPI whereas B6 Mpst transgene showed impaired PPI. SCZ postmortem brains had increased expression of MPST and CBS, a gene that codes for another enzyme involved in H₂S formation. Similarly, hair follicles in SCZ patients showed greater expression of MPST than in controls and iPSC-derived neurospheres from 22q deletion SCZ patients showed higher CBS mRNA than in controls. The authors propose a pathogenic role of "sulfide stress" in a subset of SCZ.

Response:

Thank you very much for your brilliant suggestions and favorable consideration of our manuscript.

Comment #1

The proposed role of H₂S/polysulfides in SCZ is an interesting novel idea that needs to be verified in orthogonal samples, preferably in SCZ patients that show divergent levels of PPI.

Answer

Thank you for the important suggestion. The comment means that it is preferable, (1) to validate the current results using a different sample set, and (2) to measure both PPI and condition of H₂S/polysulfides stress in the same schizophrenia patients and to see a correlation between them. We have presented MPST expression levels in hair follicle cells as a parameter for the condition of H₂S/polysulfides stress. However, we do not have any experience in measuring PPI of humans. Therefore, to respond to this comment in the revision work, we have added the examination of an independent postmortem brain sample set (BA17) (2nd sample set) (**Appendix Table S8**) and evaluated phenotypic features of schizophrenia that are associated with elevated H₂S production system. In this sample set, the MPST protein expression levels were higher in the schizophrenia than in the control group (**Fig 7A**). Interestingly, the MPST levels in schizophrenia were positively correlated with symptom severity scores (a sum of positive symptom score, negative symptom score and general score) at 3 months prior to the death rated by using the Diagnostic Instrument for Brain Studies (DIBS) (**Fig 7B**). The results suggest that patients with schizophrenia under "sulfide stress" manifest relatively severe psychotic symptoms. Here it was difficult to precisely measure H₂S/polysulfides levels in human postmortem brains, because for an accurate measurement, samples should have been quickly removed and frozen immediately after death. We have added these descriptions in **the second paragraph on p. 11** in the revised manuscript:

*To verify the idea of upregulated H₂S/polysulfides production state in schizophrenia, we examined a different postmortem brain sample set (BA17) (2nd sample set) (Ohnishi et al., 2019) (**Appendix Table S8**) and evaluated phenotypic features of schizophrenia that are associated with elevated H₂S production system. In this sample set, the MPST protein expression levels were higher in the schizophrenia than in the control groups (**Fig 7A**). Interestingly, the MPST levels in schizophrenia were positively correlated with symptom severity scores (a sum of positive symptom score, negative symptom score and general score) at 3 months prior to the death rated by using the Diagnostic Instrument for Brain Studies (DIBS) (Ohnishi et al., 2019) (**Fig 7B**). The results suggest that patients with schizophrenia under "sulfide stress" manifest relatively severe psychotic symptoms. Here it was difficult to precisely measure H₂S/polysulfides levels in human postmortem brains, because for an accurate measurement, samples should have been quickly removed and frozen immediately after death.*

Comment #2

This is a very well-designed study that examined an endophenotype of SCZ in inbred mice with contrasting PPI levels, evidence for its underlying cause is presented, i.e., increased H₂S/polysulfide formation due to high levels of Mpst.

Answer

Thank you very much for this comment.

Comment #3

Extending the investigation to SCZ-derived tissue samples, the authors found support for the potential pathogenic role of H₂S/polysulfides.

Answer

Thank you very much for this comment.

Comment #4

The authors performed extensive studies in mice, creating knockout and transgenic models in the appropriate mouse strain.

Answer

Thank you very much for this comment.

Comment #5

Postmortem brain analysis and SCZ iPSC-derived neurospheres reinforced findings generated from mouse studies.

Answer

Thank you very much for this comment. And to further validate the findings of postmortem brain study, we have analyzed a different set of postmortem brain samples as described in the answer to the comment #1, in the revision work.

Specific comment #1

In the text describe briefly the hair follicle method, indicating the number of patients and controls that donated hair follicles for the study, the number of hair follicles taken from each individual and the total RNA yield per number of hair follicles and whether amplification of RNA was performed prior to analysis.

Answer

According to this comment, we have added the following description **in the last paragraph on p. 32:**

- *We collected 10~12 hair follicles from an individual, with RNA yield per one hair follicle being 58.09 ± 27.61 ng (mean \pm SD). We did not amplify RNA but pre-amplified cDNA of all the target genes before quantification, using TaqMan PreAmp Master Mix (ThermoFisher).*

And we showed the sample numbers,

“*n* = 166 for control and *n* = 149 for schizophrenia”, **in the second paragraph on p. 12.**

Specific comment #2

Do any of the genes that encode the sulfide/sulfane forming enzymes map to the region of the top association hits for schizophrenia?

Answer

It is an important point. None of the genes that encode the sulfide/sulfane-forming enzymes map to the region of the top association hits for schizophrenia. We think that this is because epigenetic modifications of these genes contribute to risk for schizophrenia as described in the manuscript, but not inherent sequence variations of these genes.

Specific comment #3

Any possible alternative reasons for the PPI difference between B6 and C3H other than that proposed in this report?

Answer

This is also an important point. Biological traits including PPI can be determined by both genetic components and epigenetic components. Regarding the genetic components, we have previously performed a large-scale genetic study (quantitative trait loci analysis) to identify genetic underpinnings that can explain for the differences of PPI between B6 and C3H strains (Watanabe et

al., *PLoS Biology* 5: e297, 2007). We calculated a heritability of ~40% (depending on the prepulse levels), which means that ~60% of phenotypic variance can be determined by non-genetic elements including epigenetic modifications as shown in the current study. But to avoid a complicated story for readers and because this issue is out of the current scope, we have decided not to mention about the details of this issue in the manuscript.

Specific comment #4

Describe the methylation experiment more clearly, e.g., explain the source for the "CpG227" probe.

Answer

To respond to the point, we have added the following sentence as described below, in the second section on p. 40, and have prepared the new Table, **Appendix Table 17**.

*The sequence information on the probes used for DNA methylation analysis and information on the lengths of PCR products is shown in **Appendix Table 17**,*

Specific comment #5

"Sulfide content in the brain was higher in B6 mice than in C3H mice". This is the heading of the section on page 8. Is this correct? Rather, should it read "Sulfide content in the brain was higher in C3H mice than in B6 mice"?

Answer

We appreciate it very much. We have amended the heading of the section **on p. 8** in the revised manuscript.

Response to the comments of reviewer #2:

The topic of the submission is clearly interesting and it is obvious that a lot of work and state-of-the-art techniques have gone into this project. However, there are several potential issues that need to be addressed:

Response:

Thank you very much for your positive evaluation and favorable consideration of our manuscript. In the revision work, we have made every effort to answer to the concerns raised by the reviewer #2.

MAJOR -1

- The overall 'message' of the paper is unclear as far as the role of H₂S and polysulfides is concerned. In some parts, it seems that the authors regard the upregulated H₂S pathways as responses to an earlier (perhaps perinatal) oxidative stress; in other parts it seems that the authors regard these pathways as causative for schizophrenia (e.g. the fact that NaHS donation on its own can induce some features of the disease). These concepts should be better integrated into a holistic concept.

Answer:

In this study, we have aimed to elucidate the major two issues, (1) the cause (mechanistic origin) of upregulated H₂S pathway, (2) mechanistic effect (outcome) of elevated H₂S pathway, after we observed the upregulation of H₂S/polysulfides synthetic pathways in animal model of schizophrenia and human schizophrenia.

Regarding the issue of (2), our analyses are mainly summarized in the section of **"Excess H₂S/polysulfides production elicits dampened expression of genes for energy metabolism, and impairs mitochondrial energy metabolism and diminishes spine density"** (p. 13, the heading is amended in the revised manuscript). In the revision, we have strived to get better mechanistic insight into this issue in addition to the original RNA-seq data, and for this end we have analyzed energy metabolism in the *Mpst* Tg mice, and found that ATP content and ATP-to-ADP ratio are diminished in the *Mpst* Tg mice, and further mitochondrial complex IV activity is impaired in these mice (new **FIG 9A-D**). We have further obtained the data that the upregulation of *Mpst* in mice leads to reduced spine density in the hippocampus, which is reported in human study (Rosoklija G et al., *Arch Gen Psychiatry* 2000 Apr;57(4):349-56). In addition, we have detected reduced expression of Pvalb in the frontal cortex of *Mpst* Tg mice (**FIG 8D and E**), Impairments of parvalbumin (Pvalb, PV)-containing inhibitory GABA (γ -aminobutyric acid) neurons in the cortex is a well-recognized finding in schizophrenia pathophysiology (Lewis DA and Hashimoto T and Volk DW: *Nat Rev Neurosci* 6: 312-, 2005), and PV-positive GABAergic neurons are energy-demanding because of high-frequency firing (Tremblay R et al., *Neuron* 91: 260-, 2016) (**the second paragraph on p. 14**). These results suggest that an elevated H₂S/polysulfides synthetic state leads to schizophrenia onset

and progression, at least partly through dampened mitochondrial respiratory function (**in ABSTRACT, and in DISCUSSION p. 20**).

Regarding the issue of (1), our analyses are mainly summarized in the section of “**Inflammatory/oxidative stress in the early brain developmental stage results in upregulated H₂S/polysulfides production**” (p. 18). In the revision, in addition to the analysis of poly-I:C model mice, we have further consolidated the role of inflammatory/oxidative stress in the early brain developmental stage for the enhancement of H₂S/polysulfides production system in adulthood, by adding the data that inflammatory genes *Il1b*, *Il6* and *Tnf* were upregulated in C3H brain compared to B6 at some point(s) during E (embryonic) 16.5 to P (postnatal) 7 without downregulation in any periods (time points), though downregulation of *Il6* was seen at an adult stage (12-week old) (new **Fig 10J-L**) (**second paragraph on p. 18**). Here, B6 mice have shown higher H₂S/polysulfides production than C3H mice.

Based on these novel data and the reviewer’s suggestion, we have rephrased parts of ABSTRACT and DISCUSSION (**first and last paragraph on p. 22**) with respect to our holistic concept of a principle of schizophrenia genesis.

MAJOR-2

- (1) *It is unclear what enzyme (3-MST or CBS or both?) seems to play a role. In different assays (brain tissues, in vitro engineered neurospheres, hair follicle-derived materials), sometimes it is CBS, other times it is 3-MST. This appears to be confusing.*
- (2) *Also, when it comes to human tissues, it is unclear why authors did not study brain parts that are traditionally considered more relevant for schizophrenia (temporal lobe, frontal cortex?)*
- (3) *In fact in the supplemental data it is shown that the frontal cortex shows no differences in 3MST and CBS expression?*

Answer:

- (1) Regarding this issue, the detailed mechanism remains elusive. But we have speculated in the original manuscript, that “depending on the tissue-specific variability in the expression levels of these genes, a gene network including predominantly expressed one in the relevant tissue (see Fig EV3) may leverage maximum control of H₂S/polysulfides production..” (**p. 19 DISCUSSION**)
- (2) We have analyzed the BA8 region of postmortem brain (**Appendix Table S7**). This is a part of frontal cortex, and we have published many papers using this sample in our prior schizophrenia research. In the revision, we have further examined an independent sample set of postmortem brains (2nd sample set, **Appendix Table S8**) to see correlation between H₂S/polysulfides production status and clinical features before death. These samples were from BA17, because for the correlation analysis, samples of frontal cortex were not available.
- (3) We are sorry that we cannot find the “supplemental data” that the reviewer refers to.

MAJOR-3

- *The extent of the changes in 3-MST or CBS expression, in some of the experiments, appears to be minimal (perhaps a 5% change from control; Fig 6A, 6B, 6I; Fig S10H-I).*
- (1) *First of all, one should subject the data to an independent statistical review to ascertain that the analysis is correct.*
- (2) *Second, even if the difference is statistically detectable, it is questionable whether such small changes are biologically relevant in human disease.*

Answer:

- (1) One of the coauthors (Shabeesh Balan) is an expert in statistics. He has re-checked all the results and confirmed that all the data are correct in term of statistical evaluation. In addition, we have made figures more visible and understandable in the revision.
- (2) Hydrogen sulfide (H₂S) is a very critical agent in organisms. Even small changes in its levels should have serious consequences in life. Therefore, we believe that even small changes can be biologically meaningful. In the revised manuscript, we have added this point in DISCUSSION (**the first paragraph on p. 19**)

MAJOR-4

- *Some of the animal data do not fully support the authors' conclusions. For example, the difference between the two strains in ASR responses (Fig 4B) is large but the effect of 3-MST deficiency or 3-MST overexpression on the same response (Fig 4C, 4E) is minimal. These data would indicate that the majority of the factors that explain the strain difference are actually not related to the 3-MST pathway but to other factors.*

Answer:

This comment is related to the specific comment #3 of the reviewer #1. In general, effect size of each of responsible factors for complex traits like PPI and schizophrenia is very small. We have previously performed a large-scale genetic study (quantitative trait loci analysis: QTL) to identify genetic underpinnings that can explain for the differences of PPI between B6 and C3H strains (Watanabe et al., *PLoS Biology* 5: e297, 2007). In that study, we found that the chromosome-10 QTL gave an exceptionally high logarithm of Odds (LOD) score, but importantly even the chromosome-10 can explain only ~10 % of phenotypic variance, with the other significant chromosomal loci being 0.2% - 6.6% (Watanabe et al., *PLoS Biology* 5: e297, 2007). Based on our prior QTL study and other studies analyzing genetic architecture of complex traits, not-very-large effect of 3-MST should be reasonable. To avoid a complicated story for readers and because this issue is out of the current scope, we have decided not to mention about the details of this issue in the manuscript.

MAJOR-5

- No mechanism is studied or proposed (at least directly; the ingenuity pathway analysis is only correlative) as far as the actual mechanism of the excess H2S is concerned. How does the extra H2S promote the pathogenesis of schizophrenia? Much more direct experimental work would be needed to define this point.

Answer:

We hope that the reviewer can find the answer to this comment in our response to the MAJOR-1 comment of this reviewer (Regarding the issue of (2)).

MAJOR-6

- While the experiments with PolyIC are interesting (on their own), and they do demonstrate that the polyIC exposure CAN affect H2S-pathway-related pathways, this experiment does not actually prove or demonstrate that this is, indeed the case for schizophrenia patients or for the strain difference that the entire project started out with. Is it known that one strain of mice has perinatal immune stimulation or oxidative stress while the other does not?

Answer:

We hope that the reviewer can find the answer to this comment in our response to the MAJOR-1 comment of this reviewer (Regarding the issue (1)).

MINOR-1

The CPG methylation data are correlative but do not prove a mechanistic relation; in order to prove this, additional data would be needed.

Answer:

We agree that it is an important point. However, we are afraid that it would be realistically difficult to prove direct or causal link between expressional changes of target genes and changes in CpG methylation status in the CpG islands of target genes. Therefore, we have stated that, “*We further demonstrated that altered expression of these genes could be, at least in part, primed by changes in genomic DNA methylation markers, although their direct link remains to be proved.*” (the underlined portion is added in the revised manuscript (**end of the p. 19**)).

MINOR-2

If 3-MST is a H2S producing enzyme, how is it possible that plasma H2S levels are not different between WT and 3-MST KO mice or WT and 3-MST overexpressors?

Answer:

We have not measured the levels of free H₂S or polysulfides in plasma samples from mice. Meanwhile, the large proportion of Mpst protein in blood sample exists in red blood cells (RBCs) (Woldek L et al., *Acta Biochimica Poonica* 29: 121-133, 1982). Because RBCs do not have nuclei, the levels of Mpst in plasma may not be affected by differential gene expression in other tissues between different mouse strains. We have added this issue **in the second paragraph on p. 12**.

Response to the comments of reviewer #3:

The authors have already reported that, across 4 strains of mice, C57BL/6N (B6) mice exhibited the highest prepulse inhibition (PPI) scores while C3H/HeN (C3H) the lowest (Watanabe et al, 2007). Prepulse inhibition (PPI) of the startle reflex has been suggested as a candidate endophenotype for schizophrenia research, as it shows high heritability and has been found deficient in schizophrenia spectrum disorders (Roussos et al, 2016).

Furthermore, Watanabe et al., 2007 were able to identify *Fabp7* as Quantitative Trait Locus for a Schizophrenia Endophenotype. *Fabp7* (fatty acid binding protein 7, brain) is a gene with functional links to the N-methyl-D-aspartic acid (NMDA) receptor.

The manuscript of Ide et al. presents a convincing set of results, based on state-of-the-art methodologies to demonstrate that prepulse inhibition (PPI) endophenotypes levels depends on hydrogen sulfide and polysulfides production.

The manuscript is articulated on (i) proteomics studies of prefrontal cortex between the two inbred strains, (ii) the analysis of *Mpst* and genes encoding the other H₂S-producing enzymes and profile of H₂S metabolic states in mice, (iii) the study of mouse behaviors and sulfide deposition in *Mpst* KO and *Mpst*-Tg mice, (iv) the expression levels of genes encoding H₂S-producing enzymes and between DNA methylation levels and gene expression levels, (v) H₂S-producing genes/proteins in human brain-related samples, (vi) the RNA-seq analysis of frontal cortex/ *Mpst* KO and Tg mice and (vii) Gene expression in the maternal immune activation mouse model.

Altogether, these results are robust and underscore the role of an excess hydrogen sulfide and polysulfides production as able to generate a schizophrenia endophenotype.

This manuscript deserves to be published in EMBO Molecular Medicine with minor revision.

Response:

Thank you very much for valuable comments and favorable consideration of our manuscript.

MINOR-1

Abstract: abstract needs to be modified, tacking in account "The paper explained" with emphasis on "neurodevelopmental hypothesis" and analysis of a maternal immune activation mouse model for schizophrenia, that are not mentioned in the abstract.

Answer:

We have amended the Abstract according to this important comment.

MINOR-2*Introduction:*

As EMBO Molecular Medicine is not a psychiatry specialized journal, "maternal immune activation mouse model for schizophrenia" and "neurodevelopmental hypothesis of schizophrenia" need to be introduced here.

Some general references (i.e. Knuesel I, Chicha L, Britschgi M, Schobel SA, Bodmer M, Hellings JA, Toovey S, Prinssen EP. Maternal immune activation and abnormal brain development across CNS disorders. Nat Rev Neurol. 2014 Nov;10(11):643-60; Myka L, Estes and A. Kimberley McAllister. Maternal immune activation: Implications for neuropsychiatric disorders, Science 2016) need to be included here.

Answer:

We thank the reviewer for pointing out this issue. We have added details on the neurodevelopmental hypothesis of schizophrenia and maternal immune activation in the introduction section to give a brief overview on neurodevelopmental deficits in schizophrenia (**the first paragraph on p. 5**).

MINOR-3*Results:*

Excess H₂S/polysulfides production elicits dampened expression of genes for energy metabolism (p12)

*The authors indicate: "Interestingly, the dysregulated genes in the *Mpst* Tg mice were significantly enriched in the glutamatergic synaptic transmission and synaptic signaling-related ontology terms, along with the cellular metabolic processes (Fig 8C and Supplementary Table S11)". I suggest to present these results in more details as they are related to "neurodevelopmental hypothesis" and previous results on PPI, identify *Fabp7* and NMDA receptors. Similarly, "axonal guidance" pathway needs to be discussed in regard of cortical long-range projection pathways that are involved in schizophrenia.*

Answer:

We thank the reviewer for his/her valuable comments. Considering the role of *Mpst*/H₂S in regulating PPI endophenotype and neurodevelopment, we have discussed about the possible overlap of the genetic network of PPI endophenotype with that of other schizophrenia-related pathophysiology including impaired neurogenesis, NMDA signaling, and glial cell integrity (**at the end of p. 13**).

Furthermore, transcriptomic data from the *Mpst* Tg brain showed axonal guidance as one of the pathways enriched for the upregulated genes. At this moment, we do not have any validated evidence to substantiate/discuss the relevance of this observation in relation to the schizophrenia pathogenesis, within the scope of the research question addressed in the manuscript.

MINOR-4

Inflammatory/oxidative stress in the early brain developmental stage results in upregulated H₂S/polysulfides production (p15-16)

Fold changes for Sod1 and Sod2 (encoding superoxide dismutase), Cat (encoding catalase) and Gpx1 and Gpx4 (encoding glutathione peroxidase need to be indicated and discussed (Fig. 9E-I). In spite of the statistically significant changes evidences, fold-changes are quite small and this point needs to be emphasized.

Answer:

We have responded to these points **in the first paragraph on p. 18**.

MINOR-5

Discussion (p16-19)

"maternal immune activation mouse model for schizophrenia" and "neurodevelopmental hypothesis of schizophrenia " need to be discussed here.

The phenotype of Cbs overexpression mouse line that displays Novel Object recognition (NOR) defects more in relation with Intellectual Disability (ID) than schizophrenia (Marechal D, Brault V, Leon A, Martin D, Lopes Pereira P, Loaïc N, Birling MC, Friocourt G, Blondel M, Herault Y. Cbs overdosage is necessary and sufficient to induce cognitive phenotypes in mouse models of Down syndrome and interacts genetically with Dyrk1a. Hum Mol Genet. 2019 May 1;28(9):1561-1577). These novel results need to be discussed.

Answer:

We thank reviewer for his/her suggestions. We have incorporated the novel results for the trisomy of *Cbs* in different rodent models of Down syndrome, which has revealed defects in novel object recognition test (NOR). However, it is well-known, that in rodents NOR is analogous to human declarative (episodic) memory, one of the seven cognitive domains which are abnormal in schizophrenia (Curr Pharm Des. 2014; 20(31):5104-14). Hence, we cannot exclude the possibility of H₂S-mediated signaling involved in some aspect(s) of cognitive deficits manifested in schizophrenia patients (**first paragraph on p. 21**).

MINOR-6

Legends of Figures

The legends need to be improved (i.e. p. 56, Fig 9 E-I: no information is given).

Answer:

Thank you for pointing out this. We have amended them (**p. 65**).

2nd Editorial Decision

17th September 2019

Thank you for the submission of your revised manuscript to EMBO Molecular Medicine. We have now received the enclosed reports from the referees that were asked to re-assess it. As you will see the reviewers are now globally supportive and I am pleased to inform you that we will be able to accept your manuscript pending editorial following final amendments.

***** Reviewer's comments *****

Referee #2 (Remarks for Author):

The revision has addressed most of my issues. This is an important piece of work; an enormous

amount of effort has gone into it, and while there are still some open questions it is suitable for publication.

Referee #3 (Comments on Novelty/Model System for Author):

Discussed in previous version of the manuscript

Referee #3 (Remarks for Author):

The extremely precise rebuttal letter (14 pages) gave insightful responses to various points raised by the reviewers.

The modified version of Ide et al. displays great improvement as compared to previous versions of this important paper.

One can note that the authors added experiments that have been included in modified Figures 7 to 10.

Figure 7. Assessments of MPST in postmortem brains, and its association with symptom severity score and MPST in hair follicle samples as a biomarker.

Figure 8. RNA-seq and RT-PCR analyses of Mpst KO and Tg mice.

(In Figure 8: Please correct label F by D, D by E and E by F).

Figure 9. Energy metabolism and spine analyses of Mpst Tg mice.

Figure 10. Gene expression in the maternal immune activation mouse model and in multiple developmental stages

These modifications involve new set of experiments on (i) a new postmortem brain cohort, (ii) Pvalb analysis in non-Tg and Mpst Tg, as a biomarker of interneurons, (iii) Energy metabolism and spine analyses of Mpst Tg mice and (iv) addition of time points in the analysis.

I recommend publication of the manuscript, without any reservations.

2nd Revision - authors' response

25th September 2019

The Authors have addressed all Referee comments.

Corresponding Author Name: Takeo Yoshikawa

Manuscript Number: EMM-2019-10695